# New Genetic Bomb Trigger: Design, Synthesis, Molecular Dynamics Simulation, and Biological Evaluation of Novel BIBR1532-Related Analogs Targeting Telomerase against Non-Small Cell Lung Cancer

**DOI:** 10.3390/ph15040481

**Published:** 2022-04-14

**Authors:** Haytham O. Tawfik, Anwar A. El-Hamaky, Eman A. El-Bastawissy, Kirill A. Shcherbakov, Alexander V. Veselovsky, Yulia A. Gladilina, Dmitry D. Zhdanov, Mervat H. El-Hamamsy

**Affiliations:** 1Department of Pharmaceutical Chemistry, Faculty of Pharmacy, Tanta University, Tanta 31527, Egypt; anwaramin@pharm.tanta.edu.eg (A.A.E.-H.); emanesmat@pharm.tanta.edu.eg (E.A.E.-B.); mhamamsy@pharm.tanta.edu.eg (M.H.E.-H.); 2Laboratory of Medical Biotechnology, Institute of Biomedical Chemistry, Pogodinskaya St. 10/8, 119121 Moscow, Russia; kirill.soff@gmail.com (K.A.S.); veselov@ibmh.msk.su (A.V.V.); leonova_y@mail.ru (Y.A.G.); 3Department of Biochemistry, Peoples’ Friendship University of Russia (RUDN University), Miklukho-Maklaya St. 6, 117198 Moscow, Russia

**Keywords:** 2-Amino-3-cyanothiophene, BIBR1532, telomerase enzyme, TRAP assay, molecular dynamics simulation, inhibitors, lung cancer

## Abstract

Telomeres serve a critical function in cell replication and proliferation at every stage of the cell cycle. Telomerase is a ribonucleoprotein, responsible for maintaining the telomere length and chromosomal integrity of frequently dividing cells. Although it is silenced in most human somatic cells, telomere restoration occurs in cancer cells because of telomerase activation or alternative telomere lengthening. The telomerase enzyme is a universal anticancer target that is expressed in 85–95% of cancers. **BIBR1532** is a selective non-nucleoside potent telomerase inhibitor that acts by direct noncompetitive inhibition. Relying on its structural features, three different series were designed, and 30 novel compounds were synthesized and biologically evaluated as telomerase inhibitors using a telomeric repeat amplification protocol (TRAP) assay. Target compounds **29a**, **36b**, and **39b** reported the greatest inhibitory effect on telomerase enzyme with IC_50_ values of 1.7, 0.3, and 2.0 μM, respectively, while **BIBR1532** displayed IC_50_ = 0.2 μM. Compounds **29a**, **36b**, and **39b** were subsequently tested using a living-cell TRAP assay and were able to penetrate the cell membrane and inhibit telomerase inside living cancer cells. Compound **36b** was tested for cytotoxicity against 60 cancer cell lines using the NCI (USA) procedure, and the % growth was minimally impacted, indicating telomerase enzyme selectivity. To investigate the interaction of compound **36b** with the telomerase allosteric binding site, molecular docking and molecular dynamics simulations were used.

## 1. Introduction

Telomeres are unique structures composed of noncoding hexanucleotide tandem repeats 5′-(TTAGGG)_n_-3′ with 4–12 kb as a double-strand, followed by 50–400 nucleotides as a single strand overhang located at the ends of eukaryotic chromosomes [1,2,3]. Telomeres play a crucial role in the replication and proliferation of most somatic cells, as in each cell division, the telomere length gradually shortens [4]. After reaching approximately 50 cycles of cell division (Hayflick limit) [4,5], the telomere becomes too short for adopting the T-loop configuration, eliciting the DNA damage signaling pathway [1,4,6,7]. In case the linear telomere has sufficient length (7–4 kb) for binding with shelterin proteins, it will be protected from fusions by activation of p53. p53 activation resulting in cell cycle arrest at the G1 phase is known as replicative senescence or mortality stage 1 (M1). Some cells can escape from senescence by p53 suppression, resulting in further replication producing a shorter telomere. A too-short linear telomere with a critical length (2–3 kb) cannot bind with shelterin proteins, resulting in replicative crisis or mortality stage 2 (M2) [1]. By these mechanisms, telomeres function as a mitotic clock granting the cells a finite capacity for replication, and as a genetic time bomb limiting tumor growth [8,9,10,11].

Telomerase is a ribonucleoprotein complex composed of two essential components: the catalytic core subunit *hTERT* (human telomerase reverse transcriptase) and the template RNA *hTERC* (human telomerase RNA component) [12]. Telomerase is responsible for maintaining telomere length and chromosomal integrity of the frequently dividing cells [13]. Despite being silent in most human somatic cells, cancer cells undergo telomere restoration via telomerase activation or alternative telomere lengthening (ALT) [4,14,15]. Interestingly, 85–90% of human cancers are concomitant with the reactivation of the telomerase enzyme to maintain the chromosomal ends during cellular proliferation, resulting in unlimited proliferation and immortalization of tumor cells [13,16].

Telomerase activity has been identified in various types of human cancers such as breast, kidney, colon, cervix, lung, liver, pancreas, thyroid, prostate, and urinary bladder cancers [4].

Most of the chemotherapeutic drugs used are not selective to cancer cells, consequently affecting normal and healthy cells [4]. Telomerase has emerged as a potent and selective molecular target for cancer treatment that is only activated in cancerous cells [4]. Unlike growth factor receptors, the gene nonredundancy of telomerase makes tumors less likely to develop resistance to telomerase-based therapies [9,17]. A potential drawback is the delay needed for telomere consumption [18]. Therefore, telomerase inhibitors cannot be used as the first line of cancer treatment [19]. Instead, they can prevent the reactivation of cancer cells that have survived after standard treatment [19].

Lung cancer is the world’s second most common cancer and the leading cause of cancer mortality [20,21,22]. NSCLC (non-small cell lung cancer) accounts for about 84% of lung cancer cases, with telomerase activity seen in 80% [21,23]. The telomerase enzyme has been reactivated in NSCLC due to *hTERT* promoter (*hTERTp*) mutations, *hTERTp* methylation, or *hTERT* gene amplification [24,25]. Chemotherapies for advanced NSCLC are usually associated with tumor recurrences requiring additional treatments in the second- and third-line settings [26]. This recurrence is due to cancer heterogeneity and acquired resistance [24,27]. Therefore, there is an outstanding need to develop novel approaches that work across resistant types to provide patients with enduring remission [19]. Accordingly, telomerase inhibitors can be used as adjuvant therapy to block the regrowth of residual cancer cells [19].

The development or synthesis of compounds that may effectively interfere with numerous essential factors in carcinogenesis is one of the potential approaches to overcome the above mentioned disadvantages, and a number of dual-hybrid carbonic anhydrase-hTERT inhibitors have been discovered [28,29].

Telomerase targeting strategies range from immunotherapies such as the *hTERT* vaccine (GV1001) in phase III clinical trials to direct and indirect telomerase inhibitors [30]. Indirect telomerase inhibitors can be classified as G-quadruplex stabilizers and telomere-disrupting nucleoside analogs [30]. G-quadruplex stabilizers are small molecules that can stabilize telomeric G-quadruplex structures and make them unavailable for the action of telomerase, but toxicity is expected by binding to nontelomeric G-quadruplexes [30,31]. 6-Thio-dG (6-thio-2′-deoxyguanosine) is an example of a telomere-disrupting nucleoside analog that is in phase II of clinical trials (Figure 1) [30,32]. It acts as an uncapping agent by its incorporation into the telomere, impeding shelterin complex binding and resulting in the activation of the DNA damage response (DDR) [30]. Direct telomerase inhibitors can either act on *hTERC* as oligonucleotide inhibitors or on *hTERT* as small molecule inhibitors [30]. Imetelstat (GRN163L) is a lipid-conjugated thiophosphoramidate oligonucleotide that is complementary to the *hTERC* template region (Figure 1) [30]. However, in phase II clinical trials, no improvement in overall survival was evident in patients with advanced non-small cell lung cancer [30,33]. **BIBR1532** is a selective non-nucleoside potent telomerase inhibitor that acts by direct noncompetitive inhibition of the *hTERT* allosteric site (Figure 1) [34]. It decreases telomere length, obstructs cell proliferation, and cell senescence in a dose-dependent manner in vitro [35].

Unfortunately, **BIBR1532** suffers from poor pharmacokinetics and low cellular uptake, which limits its progress from preclinical to clinical trials [9,30,36]. Moreover, optimization of **BIBR1532** by the synthesis of various derivatives was not satisfactory [37]. The aim of the research project is to design and synthesize novel **BIBR1532** related analogous as inhibitors of telomerase enzyme, considering the **BIBR1532** pharmacophoric features.

## 2. Results and Discussion

### 2.1. Rational Design

The design of our target compounds was based on the pharmacophoric features deduced from the crystal complex of *Tribolium castaneum TERT* (*tcTERT*) with **BIBR1532** (PDB ID: 5CQG), in addition to valuable structure activity relationship studies (SARs) [37,38,39]. **BIBR1532** lays in a shallow, solvent-accessible, hydrophobic FVYL pocket that is conserved between *tcTERT* and *hTERT* [38]. **BIBR1532** displayed a dog bone shaped structure with two lipophilic heads separated by a four-atom linker made of an α,β-unsaturated secondary amide (Figure 2), which was essential for activity [39]. The introduction of the nitrile group is an important lead optimization approach that could enhance the ligand–receptor interaction [39]. Accordingly, 2-amino-3-cyanothiophene analogs were constructed as an advantageous lipophilic scaffold offering the amine part for our model. To enhance the lipophilicity of the amines, analog 2-amino-3-cyanocyclopenta[b]thiophene **2a** and 2-amino-3-cyano-tetrahydrobenzothiophene **2b** were prepared. Each amine was then connected to an amide linker to form an acetamide core as 2-Cyano-N-(3-cyano-4,5,6,7-tetrahydro-1-benzothiophen-2-yl)acetamide (**9b**) which has an anticancer activity [40,41,42].The second lipophilic head was designed to enclose three different series of aromatic compounds comprising monocyclic, bicyclic, and fused-ring structures (Figure 3), pointing to improved biological activity. Finally, the two lipophilic heads were connected with the essential four-atom of the α,β-unsaturated amide linker with the same geometry that mimics **BIBR1532**. Three different series were designed, and thirty novel compounds were synthesized as demonstrated in Figure 3.

### 2.2. Chemistry

A total of 30 compounds were designed and synthesized. Preparation of the anticipated amines, 2-amino-3-cyanothiophene derivatives **3a** and **3b**, was accomplished by the one-pot Gewald reaction of three components: cyclic ketone (cyclopentanone **1a** or cyclohexanone **1b**), malononitrile **2**, and elemental sulfur, using morpholine as a basic catalyst [43,44,45]. For linker construction, two subsequent reactions were employed, starting with the reaction between hydrazine hydrate (99%) **4** and ethyl cyanoacetate **5** in ethanol at 5 °C to give cyanoacetic acid hydrazide **6** [46,47], which was then reacted with acetyl acetone **7** in acidified distilled water (aq. HCl: 32%) at room temperature to yield 1-cyanoacetyl-3,5-dimethylpyrazole **8** (Figure 1) [48,49].

For effective N-cyanoacetylation of 2-amino-3-cyanothiophene derivatives **3a** and **3b**, cyanoacetic acid hydrazide **6** was converted to 1-cyanoacetyl-3,5-dimethylpyrazole **8** [48,49]. 1-Cyanoacetyl-3,5-dimethylpyrazole **8** consumed less time and produced a higher yield of products than **9a** and **9b** compared to the direct use of ethyl cyanoacetate **5** [50]. Finally, amine derivatives **3a** and **3b** were heated under reflux with 1-cyanoacetyl-3,5-dimethylpyrazole **8** in toluene to form N-cynoacetylated compounds **9a** and **9b**, while 3,5-dimethylpyrazole evolved as a highly soluble leaving group (Figure 1) [50,51].

The target compounds **25a**–**39b** were synthesized via a Knoevenagel condensation reaction to construct the α,β-unsaturated amide linker between the two lipophilic heads for series 1, 2, and 3 (Figure 2). Thereby, the aldehydes **10**–**24** were heated under the reflux for 6 h with the active methylene group containing compounds **9a**–**9b** using piperidine as a catalyst (Figure 2) [52]. To confirm whether the designed compounds were in *E* or *Z* configuration, compound **38b** was subjected to 2D NOESY NMR. If compound **38b** was in *Z* configuration, we would notice that the vinyl and nitrogen protons were close in space, unlike the *E* configuration, in which the vinyl and nitrogen protons were not close in space (Appendix A). In case of *Z* configuration, we would expect a cross peak because of coupling between nitrogen and vinyl protons by drawing a line from both 8.30 ppm (vinyl H) and 8.96 ppm (NH) signals. In our case, the absence of this cross peak at (8.30, 8.96 ppm) refers to the fact that our compound is in *E* configuration (Appendix A). Another evidence is calculation of the potential energy. The potential energy of compound **38b** was calculated by MOE 2020.9010 software and was 77.50 and 79.95 for both *E* and *Z* configurations, respectively [53]. Therefore, compound **38b** is in *E* configuration, which is more stable, and the geometry of 24 synthesized analogs is in the stable *E*-isomer form. However, compounds **25a**, **26a**, **26b**, **27a**, **27b**, and **28a** were obtained as mixtures of *E* and *Z* isomers. It was reported that telomerase inhibition is not affected by geometrical isomerism [37,39]. Resolution of geometrical isomers for compounds **26a** and **26b** was accomplished by fractional crystallization [54].

Aldehydes **14**–**16** were synthesized via an aromatic nucleophilic substitution reaction of *p-*fluorobenzaldehyde with different amines, such as piperidine, morpholine, and N-methylpiperazine (Figure 3) [55,56].

The chemical structures of the target compounds in series 1, 2, and 3 were confirmed by elemental analysis and spectroscopic data (^1^H, ^13^C NMR, and mass spectrometry) as reported in the experimental section. The ^1^H NMR spectra for compounds **9a** and **9b** were characterized by singlet signals at *δ* = 4.12 and 3.72 ppm, representing the two protons of the active methylene group, respectively. The disappearance of this signal in the ^1^H NMR spectra of compounds **25a**–**39b** is an indication of the completion of the Knoevenagel reaction, in addition to the appearance of the characteristic singlet signal of the alkenyl proton in the range of *δ* from 7.98 to 9.06 ppm. Another evidence of the completion of the Knoevenagel reaction is the 2D NOESY NMR. For compound **38b**, the vinyl proton at 8.30 ppm is coupled to both C^6^-*H* of benzodioxole ring at 7.47 ppm and C^4^-*H* of benzodioxole at 7.71 ppm (Figure 4).

The IR spectrum of compound **38b** showed the characteristic peaks at 1279.83 cm^−1^ (C-O of benzodioxole ring), 1648.26 cm^−1^ (C=O of amide), 1673.84 cm^−1^ (C=C), 2223.38 cm^−1^ (CN), 2931.53 cm^−1^ (aliphatic protons), 3002.25 cm^−1^ (aromatic protons), and 3433.83 cm^−1^ (NH of amide).

### 2.3. Biological Studies

#### 2.3.1. In Vitro Inhibition of Telomerase Enzyme

A TRAP-based assay was used to assess telomerase inhibitory activity of target compounds **25a**–**39b**. In cell-free investigations, human A549 (epithelial cell lung carcinoma) lysates were utilized to evaluate telomerase with various inhibitor concentrations. We used a highly selective telomerase and reverse transcriptase inhibitor, **BIBR1532**, as a positive control [37]. All tested compounds demonstrated dose-dependent telomerase inhibition within the range of 0.1−100 μM (Figure 5). Compounds **29a**, **36b**, and **39b** exhibited the best inhibition profile compared to the control.

The IC_50_ and IC_90_ values for **25a**–**39b** are presented in Table 1. Compounds **29a**, **36b**, and **39b** exhibited the best IC_50_ values. The most active compound, **36b**, exhibited IC_50_ = 0.3 μM. Compound **36b** revealed the strongest potency compared to **BIBR1532**, IC_50_ = 0.2 μM (Table 1).

#### 2.3.2. Living-Cell TRAP Assay

To determine whether the target compounds can affect telomerase inside living cancer cells, we incubated cancer cells A549, HCC-44, or NCI-H23 with the most potent inhibitors, **29a**, **36b**, and **39b**, and then measured telomerase activity using the TRAP assay protocol (Table 2).

Fortunately, the results indicated that all the assessed compounds suppressed telomerase in all investigated cell lines (Figure 6a,b). The most potent inhibitor was compound **36b**, which induced the most significant decrease in telomerase activity in all tested cancer cell lines. Compound **39b** demonstrated the lowest ability to inhibit telomerase up to 75.6 ± 7.9% in HCC-44 cells, while compound **29a** established moderate activity. The highest telomerase inhibition activity was observed in A549 cells. Moreover, A549 cancer cells were the most sensitive to all three tested compounds. The most active compound, **36b**, reduced telomerase activity up to 18.1 ± 5.4%. HCC-44 initially demonstrated low telomerase activity and was more resistant to inhibition (up to 54.4 ± 4.4% for **36b**). NCI-H23 demonstrated reasonable sensitivity for the tested inhibitors, and **36b** reduced telomerase up to 25.1 ± 3.2%. The results of this experiment revealed that target compounds that demonstrated significant antitelomerase activity in cell-free lysates effectively penetrated the cell membrane and inhibited telomerase inside living cancer cells.

#### 2.3.3. Telomerase Selectivity and Safety

Telomerase inhibitors exhibit a delayed onset for their cytotoxic effect to be recognized and, consequently, can be used to solve the problem of cancer relapse. The selectivity and lack of off-target effects toward other enzymes are important requirements for novel telomerase inhibitors to avoid side effects. Compound **36b** was submitted for in vitro anticancer screening to the National Cancer Institute in Bethesda, Maryland, USA [57]. NCI-60 cell line anticancer screening was implemented to ensure that compound **36b** was free from any cellular targets other than telomerase, where the incubation with cells was performed for 48 h. From the results in Table 3, we can observe that the growth percentage is barely affected by compound **36b**, suggesting the selectivity to telomerase enzyme and safety of compound **36b**.

### 2.4. In Silico Study

#### 2.4.1. Molecular Docking

A molecular docking was performed on the allosteric binding site, since our compounds have some similarity to allosteric inhibitor **BIBR1532**. The docking results showed that all compounds interact in similar poses, and the values of the scoring functions varied from −7.6 kcal/mol to −10.0 kcal/mol. Slight correlation of scoring function to the experimental inhibitory values was observed. The compounds interact with allosteric site by hydrophobic forces, hydrogen bonds, and π-cation interactions (Figure 7).

#### 2.4.2. Molecular Dynamics Simulation

To evaluate the stability of the obtained docking pose, the simulation of molecular dynamics with the most potent inhibitor **36b** was performed. The root-mean-square deviation (RMSD) values for the heavy atoms of telomerase in the complex with compound **36b** are displayed (Figure 8). The amino acid residues deviated rapidly from the initial protein structure, stabilized between 3 and 5 Å, and were still stable over the time scale of the simulation. The RMSD value for compound **36b** in the complex was also stable.

#### 2.4.3. In Silico Pharmacokinetic, Physicochemical Prediction, and PAINS Filters

SwissADME is a free web tool developed by the Swiss Institute of Bioinformatics (SIB) (http://www.swissadme.ch/, accessed on 20 December 2021) [58]. We applied SwissADME tools to predict the pharmacokinetic and physicochemical properties of the most potent inhibitor, **36b**. Compound **36b** exhibited a predicted log*P*_o/w_ = 3.04, high GIT absorption with no blood–brain barrier (BBB) permeability. Accordingly, compound **36b** has a good CNS safety profile.

The brain or intestinal estimated permeation (BOILED-Egg) model was developed by calculating both lipophilicity using the Wildman log *P* method (WLOGP) and polarity expressed in topological polar surface area (tPSA), followed by plotting the relationship between them in a BOILED-Egg diagram, as illustrated in Figure 9 [59]. Therefore, we can predict both gastrointestinal absorption and BBB permeability for the tested compound. BOILED-Egg was assembled for compounds **36b** and **BIBR1532** (Figure 9). With no BBB permeability, compound **36b** appeared in the zone of human intestinal absorption (HIA). However, **BIBR1532** exists in the BBB zone, giving a privilege to compound **36b** by avoiding CNS side effects. Moreover, both inhibitors are not P-glycoprotein substrates (negative Pgp); thus, they are not susceptible to the efflux mechanism by the Pgp transporter, which is a mechanism that emerged by some tumor cells as a drug resistance strategy (Figure 9) [60].

In addition, the bioavailability radar was constructed for inhibitors **36b** and **BIBR1532**. It is composed of six axes bearing six physicochemical properties, which are size (SIZE), lipophilicity (LIPO), solubility (INSOLU), polarity (POLAR), saturation (INSATU), and flexibility (FLEX) (Figure 10) [58]. Compound **36b** showed an improvement in fraction Csp3 (as illustrated by the INSATU property in the bioavailability radar) compared to **BIBR1532**, explaining the importance of the aliphatic portion attached to the thiophene ring. Both compounds **36b** and **BIBR1532** pass the Lipinski (Pfizer), Ghose (Amgen), Veber (GSK), and Egan (Pharmacia) filters, which are used by some pharmaceutical companies to define drug-like qualities.

SwissADME-achieved data classified compound **36b** as non-PAINS (pan-assay interference compounds), signifying the high selectivity of our target compound **36b**. It did not display a potent response in assays other than its target which is consistent with the NCI-60 cell line panel assay.

### 2.5. Structure-Activity Relationship (SAR)

Telomerase inhibitory activities for target compounds were accomplished by a modified TRAP assay [61,62,63], since **BIBR1532** served as a positive control. Thirty novel inhibitors, **25a**–**39b**, disclosed IC_50_ values ranging from 0.3 μM to 205.3 μM compared to **BIBR1532**, IC_50_ = 0.2 μM. The TRAP assay results (IC_50_ and IC_90_), recorded in Table 1, specified that we designed and synthesized three series of novel telomerase inhibitors. Sixteen compounds showed the strongest inhibitory effect, with IC_50_ values ranging between 0.3 and 13.5 μM. Seven compounds revealed a moderate inhibitory effect, with IC_50_ values ranging between 20.9 and 53.6 μM, while seven compounds displayed the weakest activity, with IC_50_ values ranging between 89.9 and 205.3 μM. Regarding the ring size of the amine part, where (*n* = 1 or 2), it was observed that target compounds having cyclopenta[b]thiophen amine, (*n* = 1), were stronger telomerase inhibitors than tetrahydrobenzothiophene (*n* = 2), analogs except for compounds **31**, **33**, **34**, **36**, and **39**. Meanwhile, the aldehyde part revealed that in series 3, inhibitors with lipophilic fused aromatic analogs, such as naphthalene in **35a**–**b** and **36a**–**b** and indole in **39a**–**b**, exhibited the best activity, with IC_50_ values ranging between 0.3 and 5.3 μM. The presence of the less lipophilic benzodioxole ring in compounds **37a**–**b** and **38a**–**b** reduced the activity, and the reported IC_50_ values ranging from 23.5 to 97.8 μM. In series 1, the presence of electron withdrawing groups, such as chloride and trifluoromethyl groups, enhanced the potency. Compounds **27a**–**b** and **28a**–**b** disclosed a dichlorobenzyl group bearing two chloride atoms either in the *ortho*, *meta,* or *meta*-*para* positions, and exhibited IC_50_ values ranging from 6.5 to 12.5 μM. Meanwhile, inhibitors **25a**–**b** with the *p*-trifluorometheylbenzyl group revealed IC_50_ values of 20.9 and 25.8 μM, respectively. In contrast, the presence of electron-donating groups reduced the potency, as revealed in compounds **26a**–**b** with *m,p*-dimethoxybenzyl group, which reported IC_50_ values of 13.5 and 53.6 μM, respectively. In series 2, having inhibitors with substituted benzaldehyde revealed that the presence of piperidine and morpholine rings in the *para* position enhanced the activity and displayed strong telomerase inhibition (**29a**–**b** and **30a**–**b**) with IC_50_ values ranging between 1.7 and 5.9 μM. The presence of the N-methylpiperazinyl group in the *para* position of the benzene ring in compounds **31a**–**b** showed a significant decrease in activity, as their polarity is not consistent with the lipophilic portion of the active site. Compounds **31a**–**b** displayed IC_50_ values of 98.5 and 89.9 μM, respectively. Moreover, compound **32b** with *m*-methoxy-*p*-benzyloxybenzyl group and *n* = 2 showed IC_50_ = 109.7 μM. Compound **32a** was an exception in the *m*-methoxy-*p*-benzyloxybenzyl group, and *n* = 1 showed IC_50_ = 6.4 μM. The presence of the polar pyrazole ring in compounds, **33a**–**b** and **34a**–**b** significantly decreased their activity (Figure 11).

## 3. Materials and Methods

### 3.1. Chemistry

Sigma-Aldrich, Alfa Aesar, and Merck provided all of the organic reagents used in this research. The open capillary method was used to determine melting points on the electrothermal melting point apparatus (Stuart SMP10), and the results were reported uncorrected. TLC was used to monitor reactions on a precoated sheet (Fastman Kodak, Silica 60 F_254_) with the following developing system: *n*-hexane:ethyl acetate (66.7:33.3) and UV light at 254 nm. The elemental analysis was conducted using a PerkinElmer 2400 CHNS analyzer (% C, H, N, and S). It was measured at Nasr City, Cairo, Egypt, at Al-Azhar University’s Regional Center for Mycology and Biotechnology. ^1^H, ^13^C NMR, and 2D NOESY spectra were recorded using CDCl_3_, DMSO-*d*_6_, or CF_3_COOD as a solvent and tetramethylsilane (TMS) as an internal reference on a Bruker FT-NMR spectrometer at 400 MHz and 100 MHz, respectively, and a JEOL ECA-500 II FT-NMR spectrometer at 500 MHz and 125 MHz. All values of chemical shift, coupling constants, *J*, and splitting [singlet (s), doublet (d), triplet (t), quintet (quint), multiplet (m), broad(br)] are expressed in ppm and Hz. The Faculty of Pharmacy at Mansoura University in Egypt and the Faculty of Science at Mansoura University in Egypt both provided ^1^H and ^13^C NMR spectra. Except for compounds **25a**, **26a**, **26b**, **27a**, **27b**, and **28a**, which had the *E*-*Z* mixture, all remaining compounds were obtained in the *E* configuration. Absence of some signals in compounds **25a**, **26b**, **30b**, **31b**, **32a**, **32b**, and **38a** was due to very low solubility. At Tanta University’s Central Laboratory, infrared (IR) spectroscopy measurements were performed at using a Bruker to record infrared spectra in the range 4000–500 cm^−1^. Thermo-Scientific ISQ Single Quadruple MS was used to record electron ionization mass spectra (EI-MS) using a 70-eV ionization energy and helium gas (carrier gas) at a constant flow rate of 1 mL/min. The mass spectrometry was performed in Nasr City, Cairo, Egypt, at Al-Azhar University’s Regional Center for Mycology and Biotechnology.

#### 3.1.1. General Procedure for the Preparation of **3a**,**b**

At room temperature, a stirred solution of suitable ketone **1a**,**b** (10 mmol) and malononitrile **2** (10 mmol) in ethanol (20 mL) was added with sulfur (10 mmol). After heating the reaction mixture to 60 °C, morpholine (12 mmol) was added dropwise and stirred for 30 min. The reaction mixture was permitted to cool and stirred for 5 h at rt. The formed precipitates **3a**,**b** were filtered and rinsed with cold MeOH and then recrystallized from EtOH [43,44,45].

*2-Amino-4H,5H,6H-cyclopenta[b]thiophene-3-carbonitrile* (**3a**). Compound **3a** was synthesized according to typical procedure using cyclopentanone **1a**. The yield was (1.31 g, 79.94%) as brown powder with m.p. 159–161 °C [43,44,45].

*2-Amino-4,5,6,7-tetrahydro-1-benzothiophene-3-carbonitrile* (**3b**). Compound **3b** was synthesized according to typical procedure using cyclohexanone **1b**. The yield was (1.49 g, 84.09%) as off-white powder with m.p. 149–151 °C [43,44,45].

#### 3.1.2. Cyanoacetic Acid Hydrazide (**6**)

Hydrazine hydrate (99%) **4** (10 mmol) was added to a solution of ethyl cyanoacetate **5** (10 mmol) in ethanol (20 mL) and stirred for 1 h at 0 °C. The formed crystals were then filtered and rinsed with cold MeOH and recrystallized from EtOH. The yield was (0.89 g, 89.91%) as colorless crystals with m.p. 114–116 °C [46,47].

#### 3.1.3. 1-Cyanoacetyl-3,5-dimethylpyrazole (**8**)

Acetyl acetone **7** (10 mmol) was added to a solution of cyanoacetic acid hydrazide **6** (10 mmol) in distilled water (20 mL) containing a catalytic amount of HCl (32%) and stirred for 1 h at rt. The formed precipitate was filtered and washed with water. The yield was (1.38 g, 84.99%) as white powder with m.p. 124–126 °C [48,49].

#### 3.1.4. General Procedure for the Preparation of **9a**,**b**

1-Cyanoacetyl-3,5-dimethylpyrazole **8** (1.2 mmol), and either **3a** or **3b** (1 mmol) in dry toluene (20 mL) were heated under reflux for 6 h [50,51]. Filtered precipitates **9a**,**b** were washed in diethyl ether before being recrystallized from EtOH.

*2-Cyano-N-{3-cyano-4H,5H,6H-cyclopenta[b]thiophen-2-yl}acetamide* (**9a**). Compound **9a** was synthesized according to typical procedure using **3a** as a starting material. The yield was (0.15 g, 64.85%) as brown powder with m.p. 239–241 °C. ^1^H NMR (400 MHz, DMSO-*d*_6_) δ (ppm): 2.37 (2H, quint., *J* = 8.0 Hz, CH_2_ of C^5^-*H*s of cyclopenta[b]thiophen ring), 2.74 (2H, t, *J* = 8.0 Hz, CH_2_ of C^4^-*H*s of cyclopenta[b]thiophen ring), 2.85 (2H, t, *J* = 8.0 Hz, CH_2_ of C^6^-*H*s of cyclopenta[b]thiophen ring), 4.12 (2H, s, CH_2_), 11.96 (1H, s, NH).

*2-Cyano-N-(3-cyano-4,5,6,7-tetrahydro-1-benzothiophen-2-yl)acetamide* (**9b**). Compound **9b** was synthesized according to typical procedure using **3b** as a starting material. The yield was (0.17 g, 69.71%) as grayish-white precipitate with m.p. 244–246 °C [50,51]. ^1^H NMR (400 MHz, CDCl_3_) δ (ppm): 1.80–2.10 (4H, m, 2 CH_2_ of C^5,6^-*H*s of tetrahydrobenzothiophene ring), 2.60–2.75 (4H, m, 2 CH_2_ of C^4,7^-*H*s of tetrahydrobenzothiophene ring), 3.72 (2H, s, CH_2_), 9.85 (1H, s, NH).

#### 3.1.5. General Procedure for the Preparation of **14**–**16**

Suitable aliphatic cyclic amines piperidine, morpholine, or N-methyl piperazine (15 mmol), and *p*-fluorobenzaldehyde (15 mmol) in DMSO (30 mL) were heated under reflux in the presence of potassium carbonate anhydrous (16 mmol) as a base for 6 h [55,56]. After pouring the reaction mixture into distilled water (30 mL), ethyl acetate was used to extract it (3 × 30 mL). The extracts were collected, brine-washed, dried over anhydrous MgSO4, and then evaporated in vacuo.

*4-(Piperidin-1-yl)benzaldehyde* (**14**). Compound **14** was synthesized according to typical procedure using piperidine as a starting material. The yield was (1.99 g, 69.99%) as a pink powder with m.p. 63–65 °C [55].

*4-(Morpholin-4-yl)benzaldehyde* (**15**). Compound **15** was synthesized according to typical procedure using morpholine as a starting material. The yield was (2.09 g, 72.96%) as a yellow powder with m.p. 65–67 °C [55].

*4-(4-Methylpiperazin-1-yl)benzaldehyde* (**16**). Compound **16** was synthesized according to typical procedure using N-methyl piperazine as a starting material. The yield was (2.05 g, 66.96%) as a yellow powder with m.p. 56–58 °C [55].

#### 3.1.6. General Procedure for the Preparation of **25a**–**39b**

In CH_3_CN (10 mL), a suitable aldehyde **10**–**24** (1 mmol), and either **9a** or **9b** (1 mmol) were heated under reflux with piperidine (three drops) as a catalyst [53]. The **25a**–**39b** precipitate was filtered and washed in acetone before being recrystallized from EtOH.

*(2E/2Z)-2-Cyano-N-{3-cyano-4H,5H,6H-cyclopenta[b]thiophen-2-yl}-3-[4-(trifluoromethyl) phenyl]prop-2-enamide* (**25a**). Compound **25a** was synthesized according to the typical procedure using **9a** and 4-(trifluoromethyl)benzaldehyde **10** as starting materials. The yield was (0.25 g, 64.53%) as off-white powder with m.p. 354–356 °C. ^1^H NMR (500 MHz, CF_3_COOD) δ (ppm): 2.65–2.75 (2H, m, CH_2_ of C^5^-*H*s of cyclopenta[b]thiophen ring), 3.08–3.14 (2H, t, *J* = 7.0 Hz, CH_2_ of C^4^-*H*s of cyclopenta[b]thiophen ring), 3.14–3.23 (2H, t, *J* = 7.0 Hz, CH_2_ of C^6^-*H*s of cyclopenta[b]thiophen ring), 7.65 (2H, d, *J* = 8.0 Hz, 2 CH of C^2,6^-*H*s of phenyl ring), 7.85 (2H, d, *J* = 8.0 Hz, 2 CH of C^3,5^-*H*s of phenyl ring), 8.30 (1H, s, vinyl-*H*), 8.53 (1H, s, NH). ^13^C NMR (125 MHz, CF_3_COOD) δ (ppm): 29.76, 30.59, 30.90, 103.02, 115.41, 128.73 (2C), 131.53 (2C), 140.22, 149.91, 153.08, 159.37 (2C), 171.17, 184.12. EI-MS: *m*/*z*: 387.39 [M^+^]. Anal. Calcd. for C_19_H_12_F_3_N_3_OS: C, 58.91; H, 3.12; N, 10.85; S, 8.28. Found: C, 59.17; H, 3.29; N, 11.08; S, 8.37.

*(2E)-2-Cyano-N-(3-cyano-4,5,6,7-tetrahydro-1-benzothiophen-2-yl)-3-[4-(trifluoromethyl)phenyl]prop-2-enamide* (**25b**). Compound **25b** was synthesized according to the typical procedure using **9b** and 4-(trifluoromethyl)benzaldehyde **10** as starting materials. The yield was (0.22 g, 54.56%) as a yellow powder with m.p. 225–227 °C. ^1^H NMR (500 MHz, DMSO-*d*_6_) δ (ppm): 1.70–1.85 (4H, m, 2 CH_2_ of C^5,6^-*H*s of tetrahydrobenzothiophene ring), 2.50–2.55 (2H, m, CH_2_ of C^7^-*H*s of tetrahydrobenzothiophene ring), 2.60–2.65 (2H, m, CH_2_ of C^4^-*H*s of tetrahydrobenzothiophene ring), 7.97 (2H, d, *J* = 8.5 Hz, CH of C^2,6^-*H*s of phenyl ring), 8.15 (2H, d, *J* = 8.5 Hz, CH of C^3,5^-*H*s of phenyl ring), 8.45 (1H, s, vinyl-*H*), 12.03 (1H, s, NH). ^13^C NMR (125 MHz, DMSO-*d*_6_) δ (ppm): 21.43, 22.33, 23.85, 24.56, 94.41, 104.80, 114.98, 117.42, 125.42, 126.21 (2C), 127.47 (2C), 133.37, 134.05, 136.50, 145.08, 148.00, 157.40, 157.91. EI-MS: *m*/*z*: 401.08 [M^+^]. Anal. Calcd. for C_20_H_14_F_3_N_3_OS: C, 59.84; H, 3.52; N, 10.47; S, 7.99. Found: C, 59.70; H, 3.72; N, 10.68; S, 8.15.

*(2E)-2-Cyano-N-{3-cyano-4H,5H,6H-cyclopenta[b]thiophen-2-yl}-3-(3,4-dimethoxyphenyl)prop-2-enamide* (**26a**). Compound **26a** was synthesized according to typical procedure using **9a** and 3,4-dimethoxybenzaldehyde **11** as starting materials. In fractional crystallization, compound **26a** was subjected to hot dioxane, where the *E* isomer was dissolved, leaving the *Z* isomer insoluble. The hot dioxane was filtered to remove the insoluble *Z* isomer and then allowed to cool to give the *E* isomer. The yield was (0.26 g, 68.52%) as a yellow powder with m.p. 249–251 °C. ^1^H NMR (500 MHz, DMSO-*d*_6_) δ (ppm): 2.30–2.45 (2H, m, CH_2_ of C^5^-*H*s of cyclopenta[b]thiophen ring), 2.60–2.95 (4H, m, 2 CH_2_ of C^4,6^-*H*s of cyclopenta[b]thiophen ring), 3.81 (3H, s, CH_3_ of OCH_3_ at C^3^ of phenyl ring), 3.87 (3H, s, CH_3_ of OCH_3_ at C^4^ of phenyl ring), 7.19 (1H, d, *J* = 7.5 Hz, CH of C^5^-*H* of phenyl ring), 7.68 (1H, d, *J* = 7.5 Hz, CH of C^6^-*H* of phenyl ring) 7.73 (1H, s, CH of C^2^-*H* of phenyl ring), 8.26 (1H, s, vinyl-*H*), 11.72 (1H, s, NH). ^13^C NMR (125 MHz, CF_3_COOD) δ (ppm): 29.24, 29.32, 30.77. 57.12 (2C), 96.92, 109.67, 113.22, 113.93, 117.25, 126.63, 132.36, 140.10, 150.50, 152.88, 154.28, 155.00, 156.87, 157.08, 171.60. EI-MS: *m*/*z*: 379.2 [M^+^]. Anal. Calcd. for C_20_H_17_N_3_O_3_S: C, 63.31; H, 4.52; N, 11.07; S, 8.45. Found: C, 63.49; H, 4.68; N, 11.29; S, 8.52.

*(2E)-2-Cyano-N-(3-cyano-4,5,6,7-tetrahydro-1-benzothiophen-2-yl)-3-(3,4-dimethoxyphenyl)prop-2-enamide* (**26b**). Compound **26b** was synthesized according to typical procedure using **9b** and 3,4-dimethoxybenzaldehyde **11** as starting materials. In fractional crystallization, compound **26b** was subjected to hot dioxane, where the *E* isomer was dissolved, leaving the *Z* isomer insoluble. The hot dioxane was filtered to remove the insoluble *Z* isomer and then allowed to cool to give the *E* isomer. The yield was (0.25 g, 62.78%) as a yellow powder with m.p. 264–266 °C. ^1^H NMR (500 MHz, DMSO-*d*_6_) δ (ppm): 1.70–1.80 (4H, m, 2 CH_2_ of C^5,6^-*H*s of tetrahydrobenzothiophene ring), 2.50–2.55 (2H, m, CH_2_ of C^7^-*H*s of tetrahydrobenzothiophene ring), 2.60–2.65 (2H, m, CH_2_ of C^4^-*H*s of tetrahydrobenzothiophene ring), 3.81 (3H, s, CH_3_ of OCH_3_ at C^3^ of phenyl ring), 3.87 (3H, s, CH_3_ of OCH_3_ at C^4^ of phenyl ring), 7.19 (1H, d, *J* = 9.0 Hz, CH of C^5^-*H* of phenyl ring), 7.68 (1H, d, *J* = 9.0 Hz, CH of C^6^-*H* of phenyl ring) 7.73 (1H, s, CH of C^2^-*H* of phenyl ring), 8.26 (1H, s, vinyl-*H*), 11.74 (1H, s, NH). ^13^C NMR (125 MHz, DMSO-*d*_6_) δ (ppm): 21.67, 22.56, 23.48, 23.60, 55.53, 55.91, 111.98, 112.59 (2C), 124.24, 126.27, 148.73 (2C), 153.19 (2C), 160.67. EI-MS: *m*/*z*: 393.64 [M^+^]. Anal. Calcd. for C_21_H_19_N_3_O_3_S: C, 64.11; H, 4.87; N, 10.68; S, 8.15. Found: C, 64.38; H, 4.92; N, 10.85; S, 8.34.

*(2E/2Z)-2-Cyano-N-{3-cyano-4H,5H,6H-cyclopenta[b]thiophen-2-yl}-3-(3,4-dichlorophenyl)prop-2-enamide* (**27a**). Compound **27a** was synthesized according to the typical procedure using **9a** and 3,4-dichlorobenzaldehyde **12** as starting materials. The yield was (0.21 g, 54.09%) as a gray powder with m.p. 329–331 °C. ^1^H NMR (500 MHz, CF_3_COOD) δ (ppm): 2.70–2.80 (2H, m, CH_2_ of C^5^-*H*s of cyclopenta[b]thiophen ring), 3.10–3.25 (4H, m, 2 CH_2_ of C^4,6^-*H*s of cyclopenta[b]thiophen ring), 7.38 (1H, s, CH of C^5^-*H* of phenyl ring), 7.60–7.85 (2H, m, 2 CH of C^2,6^-*H* of phenyl ring), 8.32 (1H, s, vinyl-*H*), 8.65 (1H, s, NH). ^13^C NMR (125 MHz, CF_3_COOD) δ (ppm): 29.65, 30.49, 30.81, 103.10, 112.48, 112.71, 115.33, 129.62, 132.58, 133.66, 136.67, 139.37, 140.09, 144.09, 149.80, 152.58, 158.01, 158.94. EI-MS: *m*/*z*: 388.5 [M^+^]. Anal. Calcd. for C_18_H_11_Cl_2_N_3_OS: C, 55.68; H, 2.86; N, 10.82; S, 8.26. Found: C, 55.87; H, 3.15; N, 11.04; S, 8.42.

*(2E/2Z)-2-Cyano-N-(3-cyano-4,5,6,7-tetrahydro-1-benzothiophen-2-yl)-3-(3,4-dichlorophenyl)prop-2-enamide* (**27b**). Compound **27b** was synthesized according to typical procedure using **9b** and 3,4-dichlorobenzaldehyde **12** as starting materials. The yield was (0.23 g, 57.42%) as a gray powder with m.p. 376–378 °C. ^1^H NMR (500 MHz, CF_3_COOD) δ (ppm): 1.85–1.95 (4H, m, 2 CH_2_ of C^5,6^-*H*s of tetrahydrobenzothiophene ring), 2.55–2.65 (2H, m, CH_2_ of C^7^-*H*s of tetrahydrobenzothiophene ring), 2.70–2.80 (2H, m, CH_2_ of C^4^-*H*s of tetrahydrobenzothiophene ring), 7.39 (1H, d, *J* = 7.0 Hz, CH of C^5^-*H* of phenyl ring), 7.67 (2H, m, 2 CH of C^2,6^-*H* of phenyl ring), 8.56 (1H, s, vinyl-*H*), 8.88 (1H, s, NH). ^13^C NMR (125 MHz, CF_3_COOD) δ (ppm): 22.89, 25.71, 26.19, 27.39, 114.72, 115.50, 116.97, 117.80, 129.53, 130.16, 132.00, 132.47, 133.57, 134.77, 136.56, 145.67, 146.33, 158.30, 158.90. EI-MS: *m*/*z*: 402.98 [M^+^]. Anal. Calcd. for C_19_H_13_Cl_2_N_3_OS: C, 56.73; H, 3.26; N, 10.45; S, 7.97. Found: C, 56.81; H, 3.42; N, 10.69; S, 7.92.

*(2E/2Z)-2-Cyano-N-{3-cyano-4H,5H,6H-cyclopenta[b]thiophen-2-yl}-3-(2,3-dichlorophenyl)prop-2-enamide* (**28a**). Compound **28a** was synthesized according to typical procedure using **9a** and 2,3-dichlorobenzaldehyde **13** as starting materials. The yield was (0.27 g, 70.57%) as a gray powder with m.p. 302–304 °C. ^1^H NMR (500 MHz, DMSO-*d*_6_) δ (ppm): 2.37 (2H, quint, *J* = 7.0 Hz, CH_2_ of C^5^-*H*s of cyclopenta[b]thiophen ring), 2.76 (2H, t, *J* = 7.0 Hz, CH_2_ of C^4^-*H*s of cyclopenta[b]thiophen ring), 2.87 (2H, t, *J* = 7.0 Hz, CH_2_ of C^6^-*H*s of cyclopenta[b]thiophen ring), 7.61 (1H, t, *J* = 8.0 Hz, CH of C^5^-*H* of phenyl ring), 7.80 (1H, d, *J* = 8.0 Hz, CH of C^6^-*H* of phenyl ring), 7.94 (1H, d, *J* = 8.0 Hz, CH of C^4^-*H* of phenyl ring), 8.48 (1H, s, vinyl-*H*), 11.99 (1H, s, NH). ^13^C NMR (125 MHz, DMSO-*d*_6_) δ (ppm): 27.48, 27.67, 29.27, 93.08, 110.58, 114.31, 114.68, 128.69, 128.97, 131.85, 132.81, 133.55, 136.11, 141.98, 149.26 (2C), 159.11 (2C). EI-MS: *m*/*z*: 388.33 [M^+^]. Anal. Calcd. for C_18_H_11_Cl_2_N_3_OS: C, 55.68; H, 2.86; N, 10.82; S, 8.26. Found: C, 55.79; H, 3.04; N, 11.03; S, 8.44.

*(2E)-2-Cyano-N-(3-cyano-4,5,6,7-tetrahydro-1-benzothiophen-2-yl)-3-(2,3-dichlorophenyl)prop-2-enamide* (**28b**). Compound **28b** was synthesized according to typical procedure using **9b** and 2,3-dichlorobenzaldehyde **13** as starting materials. The yield was (0.31 g, 77.06%) as a dark yellow powder with m.p. 232–234 °C. ^1^H NMR (500 MHz, DMSO-*d*_6_) δ (ppm): 1.70–1.80 (4H, m, 2 CH_2_ of C^5,6^-*H*s of tetrahydrobenzothiophene ring), 2.50–2.55 (2H, m, CH_2_ of C^7^-*H*s of tetrahydrobenzothiophene ring), 2.60–2.65 (2H, m, CH_2_ of C^4^-*H*s of tetrahydrobenzothiophene ring), 7.61 (1H, t, *J* = 8.0 Hz, CH of C^5^-*H* of phenyl ring), 7.87 (1H, d, *J* = 8.0 Hz, CH of C^6^-*H* of phenyl ring), 7.94 (1H, d, *J* = 8.0 Hz, CH of C^4^-*H* of phenyl ring), 8.48 (1H, s, vinyl-*H*), 12.02 (1H, s, NH). ^13^C NMR (125 MHz, DMSO-*d*_6_) δ (ppm): 21.62, 22.53, 23.50, 23.59, 96.90, 110.64, 114.00, 114.70, 128.65, 128.94, 130.00, 131.83 (2C), 132.79, 133.00, 149.18 (2C), 159.10 (2C). EI-MS: *m*/*z*: 402.54 [M^+^]. Anal. Calcd. for C_19_H_13_Cl_2_N_3_OS: C, 56.73; H, 3.26; N, 10.45; S, 7.97. Found: C, 56.85; H, 3.44; N, 10.78; S, 7.89.

*(2E)-2-Cyano-N-{3-cyano-4H,5H,6H-cyclopenta[b]thiophen-2-yl}-3-[4-(piperidin-1-yl)phenyl]prop-2-enamide* (**29a**). Compound **29a** was synthesized according to typical procedure using **9a** and 4-(piperidin-1-yl)benzaldehyde **14** as starting materials. The yield was (0.28 g, 70.55%) as a dark red powder with m.p. 262–264 °C. ^1^H NMR (500 MHz, DMSO-*d*_6_) δ (ppm): 1.50–1.65 (6H, m, 3 CH_2_ of C^3,4,5^-*H*s of piperdine ring), 2.30–2.40 (2H, quint, *J* = 6.5 Hz, CH_2_ of C^5^-*H*s of cyclopenta[b]thiophen ring), 2.70–2.80 (2H, t, *J* = 6.5 Hz, CH_2_ of C^4^-*H*s of cyclopenta[b]thiophen ring), 2.80–2.90 (2H, t, *J* = 6.5 Hz, CH_2_ of C^6^-*H*s of cyclopenta[b]thiophen ring), 3.45–3.55 (4H, m, 2 CH_2_ of C^2,6^-*H*s of the piperdine ring), 7.04 (2H, d, *J* = 8.0 Hz, 2 CH of C^3,5^-*H*s of phenyl ring), 7.91 (2H, d, *J* = 8.0 Hz, 2 CH of C^2,6^-*H*s of phenyl ring), 8.10 (1H, s, vinyl-*H*), 11.45 (1H, s, NH). ^13^C NMR (125 MHz, DMSO-*d*_6_) δ (ppm): 23.96, 25.00 (2C), 27.45, 27.64, 29.17, 47.40 (2C), 94.93, 113.21 (2C), 117.67, 119.18, 133.65 (2C), 141.55, 152.52, 153.57, 161.32. EI-MS: *m*/*z*: 402.67 [M^+^]. Anal. Calcd. for C_23_H_22_N_4_OS: C, 68.63; H, 5.51; N, 13.92; S, 7.96. Found: C, 68.80; H, 5.62; N, 14.20; S, 8.05.

*(2E)-2-Cyano-N-(3-cyano-4,5,6,7-tetrahydro-1-benzothiophen-2-yl)-3-[4-(piperidin-1-yl)phenyl]prop-2-enamide* (**29b**). Compound **29b** was synthesized according to typical procedure using **9b** and 4-(piperidin-1-yl)benzaldehyde **14** as starting materials. The yield was (0.28 g, 66.26%) as an orange powder with m.p. 262–264 °C. ^1^H NMR (500 MHz, DMSO-*d*_6_) δ (ppm): 1.55–1.65 (6H, m, 3 CH_2_ of C^3,4,5^-*H*s of piperdine ring), 1.70–1.80 (4H, m, 2 CH_2_ of C^5,6^-*H*s of tetrahydrobenzothiophene ring), 2.45–2.55 (2H, m, CH_2_ of C^7^-*H*s of tetrahydrobenzothiophene ring), 2.60–2.65 (2H, m, CH_2_ of C^4^-*H*s of tetrahydrobenzothiophene ring), 3.49 (4H, t, *J* = 5.0 Hz, 2 CH_2_ of C^2,6^-*H*s of piperdine ring), 7.06 (2H, d, *J* = 8.5 Hz, 2 CH of C^3,5^-*H*s of phenyl ring), 7.91 (2H, d, *J* = 8.5 Hz, 2 CH of C^2,6^-*H*s of phenyl ring), 8.10 (1H, s, vinyl-*H*), 11.43 (1H, s, NH). ^13^C NMR (125 MHz, DMSO-*d*_6_) δ (ppm): 21.68, 22.57, 23.45, 23.59, 23.99 (2C), 25.02 (2C), 47.40, 94.98, 95.80, 113.22 (2C), 114.20, 117.72, 119.18, 128.80, 131.39, 133.67 (2C), 146.24, 152.56, 153.58, 161.29. EI-MS: *m*/*z*: 416.22 [M^+^]. Anal. Calcd. for C_24_H_24_N_4_OS: C, 69.20; H, 5.81; N, 13.45; S, 7.70. Found: C, 69.43; H, 5.96; N, 13.72; S, 7.81.

*(2E)-2-Cyano-N-{3-cyano-4H,5H,6H-cyclopenta[b]thiophen-2-yl}-3-[4-(morpholin-4-yl)phenyl]prop-2-enamide* (**30a**). Compound **30a** was synthesized according to the typical procedure using **9a** and 4-(morpholin-4-yl)benzaldehyde **15** as starting materials. The yield was (0.28 g, 69.96%) as an orange powder with m.p. 296–298 °C. ^1^H NMR (500 MHz, DMSO-*d*_6_) δ (ppm): 2.35 (2H, quint, *J* = 6.5 Hz, CH_2_ of C^5^-*H*s of cyclopenta[b]thiophen ring), 2.74 (2H, t, *J* = 6.5 Hz, CH_2_ of C^4^-*H*s of cyclopenta[b]thiophen ring), 2.85 (2H, t, *J* = 6.5 Hz, CH_2_ of C^6^-*H*s of cyclopenta[b]thiophen ring), 3.35–3.45 (4H, m, 2 CH_2_ of C^3,5^-*H*s of morpholine ring), 3.65–3.75 (4H, m, 2 CH_2_ of C^2,6^-*H*s of morpholine ring), 7.09 (2H, d, *J* = 9.0 Hz, 2 CH of C^3,5^-*H*s of phenyl ring), 7.95 (2H, d, *J* = 9.0 Hz, 2 CH of C^2,6^-*H*s of phenyl ring), 8.15 (1H, s, vinyl-*H*), 11.54 (1H, s, NH). ^13^C NMR (125 MHz, DMSO-*d*_6_) δ (ppm): 27.45, 27.64, 29.18, 46.30 (2C), 65.74 (2C), 92.11, 96.65, 113.42 (2C), 114.47, 117.36, 120.56, 133.22 (2C), 135.25, 141.60, 152.65, 153.90 (2C), 161.13. EI-MS: *m*/*z*: 404.62 [M^+^]. Anal. Calcd. for C_22_H_20_N_4_O_2_S: C, 65.33; H, 4.98; N, 13.85; S, 7.93. Found: C, 65.47; H, 5.12; N, 14.09; S, 7.81.

*(2E)-2-Cyano-N-(3-cyano-4,5,6,7-tetrahydro-1-benzothiophen-2-yl)-3-[4-(morpholin-4-yl)phenyl]prop-2-enamide* (**30b**). Compound **30b** was synthesized according to typical procedure using **9b** and 4-(morpholin-4-yl)benzaldehyde **15** as starting materials. The yield was (0.29 g, 69.05%) as an orange powder with m.p. 246–248 °C. ^1^H NMR (500 MHz, DMSO-*d*_6_) δ (ppm): 1.70–1.82 (4H, m, 2 CH_2_ of C^5,6^-*H*s of tetrahydrobenzothiophene ring), 2.49–2.54 (2H, m, CH_2_ of C^7^-*H*s of tetrahydrobenzothiophene ring), 2.59–2.64 (2H, m, CH_2_ of C^4^-*H*s of tetrahydrobenzothiophene ring), 3.40 (4H, t, *J* = 5.0 Hz, 2 CH_2_ of C^3,5^-*H*s of morpholine ring), 3.72 (4H, t, *J* = 5.0 Hz, 2 CH_2_ of C^2,6^-*H*s of morpholine ring), 7.10 (2H, d, *J* = 9.0 Hz, 2 CH of C^3,5^-*H*s of phenyl ring), 7.95 (2H, d, *J* = 9.0 Hz, 2 CH of C^2,6^-*H*s of phenyl ring), 8.14 (1H, s, vinyl-*H*), 11.52 (1H, s, NH). ^13^C NMR (125 MHz, DMSO-*d*_6_) δ (ppm): 21.71, 22.60, 23.48, 23.60, 46.31 (2C), 65.77 (2C), 113.44 (2C), 114.30, 117.48, 120.64, 126.82, 130.60, 131.37, 133.20 (2C), 146.70, 153.88, 161.19. EI-MS: *m*/*z*: 418.81 [M^+^]. Anal. Calcd. for C_23_H_22_N_4_O_2_S: C, 66.01; H, 5.30; N, 13.39; S, 7.66. Found: C, 66.28; H, 5.47; N, 13.58; S, 7.85.

*(2E)-2-Cyano-N-{3-cyano-4H,5H,6H-cyclopenta[b]thiophen-2-yl}-3-[4-(4-methylpiperazin-1-yl)phenyl]prop-2-enamide* (**31a**). Compound **31a** was synthesized according to typical procedure using **9a** and 4-(4-methylpiperazin-1-yl)benzaldehyde **16** as starting materials. The yield was (0.29 g, 69.45%) as a red powder with m.p. 254–256 °C. ^1^H NMR (500 MHz, MHz, DMSO-*d*_6_) δ (ppm): 2.25–2.35 (2H, m, CH_2_ of C^5^-*H*s of cyclopenta[b]thiophen ring), 2.44 (3H, s, CH_3_ of N-methyl of piperazine ring), 2.60–2.85 (8H, m, 2 CH_2_ of C^4,6^-*H*s of cyclopenta[b]thiophen ring and 2 CH_2_ of C^3,5^-*H*s of piperazine ring), 3.40–3.55 (4H, m, 2 CH_2_ of C^2,6^-*H*s of piperazine ring), 7.06 (2H, d, *J* = 7.5 Hz, 2 CH of C^3,5^-*H*s of phenyl ring), 7.87 (2H, d, *J* = 7.5 Hz, 2 CH of C^2,6^-*H*s of phenyl ring), 7.98 (1H, s, vinyl-*H*), 10.39 (1H, br s, NH). ^13^C NMR (125 MHz, CF_3_COOD) δ (ppm): 29.20, 29.31, 30.76, 45.25, 50.40 (2C), 54.37 (2C), 100.28, 109.91, 117.14, 120.24, 130.83, 135.88 (2C), 140.28 (2C), 149.49, 150.87, 151.99, 154.82, 163.01, 164.10. EI-MS: *m*/*z*: 417.73 [M^+^]. Anal. Calcd. for C_23_H_23_N_5_OS: C, 66.16; H, 5.55; N, 16.77; S, 7.68. Found: C, 66.34; H, 5.67; N, 16.89; S, 7.76.

*(2E)-2-Cyano-N-(3-cyano-4,5,6,7-tetrahydro-1-benzothiophen-2-yl)-3-[4-(4-methylpiperazin-1-yl)phenyl]prop-2-enamide* (**31b**). Compound **31b** was synthesized according to typical procedure using **9b** and 4-(4-methylpiperazin-1-yl)benzaldehyde **16** as starting materials. The yield was (0.32 g, 74.14%) as a golden powder with m.p. 257–259 °C. ^1^H NMR (500 MHz, CF_3_COOD) δ (ppm): 2.00–2.10 (4H, m, 2 CH_2_ of C^3,5^-*H*s of piperazine ring), 2.95–3.15 (4H, m, 2 CH_2_ of C^5,6^-*H*s of tetrahydrobenzothiophene ring), 3.21 (3H, s, CH_3_ of N-methyl of piperazine ring), 3.70–3.80 (2H, m, CH_2_ of C^7^-*H*s of tetrahydrobenzothiophene ring), 4.05–4.15 (4H, m, 2 CH_2_ of C^2,6^-*H*s of piperazine ring), 4.30–4.35 (2H, m, CH_2_ of C^4^-*H*s of tetrahydrobenzothiophene ring), 7.68 (2H, d, *J* = 9.5 Hz, 2 CH of C^3,5^-*H*s of phenyl ring), 8.29 (2H, d, *J* = 9.5 Hz, 2 CH of C^2,6^-*H*s of phenyl ring), 8.71 (1H, s, vinyl-*H*), 9.39 (1H, s, NH), 8.91 (1H, s, NH_4_^+^ of piperazine ring as a result of ionization). ^13^C NMR (125 MHz, CF_3_COOD) δ (ppm): 20.72, 22.64, 23.11, 26.58, 45.41, 49.44 (2C), 54.84 (2C), 98.84, 115.15, 117.50, 121.69, 131.41, 135.38, 136.29 (2C), 145.36 (2C), 151.06, 152.93, 155.62, 167.26. EI-MS: *m*/*z*: 431.25 [M^+^]. Anal. Calcd. for C_24_H_25_N_5_OS: C, 66.80; H, 5.84; N, 16.23; S, 7.43. Found: C, 67.04; H, 5.98; N, 16.48; S, 7.60.

*(2E)-3-[4-(Benzyloxy)-3-methoxyphenyl]-2-cyano-N-{3-cyano-4H,5H,6H-cyclopenta[b]thiophen-2-yl}prop-2-enamide* (**32a**). Compound **32a** was synthesized according to typical procedure using **9a** and 4-(benzyloxy)-3-methoxybenzaldehyde **17** as starting materials. The yield was (0.22 g, 48.30%) as a yellow powder with m.p. 233–235 °C. ^1^H NMR (500 MHz, pyridine-*d*_5_) δ (ppm): 2.05–2.15 (2H, m, CH_2_ of C^5^-*H*s of cyclopenta[b]thiophen ring), 2.55–2.75 (4H, m, 2 CH_2_ of C^4,6^-*H*s of cyclopenta[b]thiophen ring), 3.78 (3H, s, CH_3_ of OCH_3_ at C^3^ of phenyl ring), 5.21 (2H, s, benzylic CH_2_), 7.08 (1H, d, *J* = 9.0 Hz, C^5^-*H* of phenyl ring), 7.22 (1H, hindered by solvent peak, C^4^-*H* of benzyl group), 7.32 (2H, d, *J* = 6.5 Hz, C^2,6^-*H*s of benzyl group), 7.38 (2H, t, *J* = 6.5 Hz, C^3,5^-*H*s of benzyl group), 7.58 (1H, hindered by solvent peak, C^6^-*H* of phenyl ring), 7.80 (1H, s, C^2^-*H* of phenyl ring), 8.53 (1H, s, vinyl-*H*). ^13^C NMR (125 MHz, pyridine-*d*_5_) δ (ppm): 27.93, 28.05, 29.49, 55.68, 70.79, 93.85, 101.89, 112.84, 113.30, 127.02, 128.10 (2C), 128.46, 128.90 (2C), 153.49, 161.37. EI-MS: *m*/*z*: 455.3 [M^+^]. Anal. Calcd. for C_26_H_21_N_3_O_3_S: C, 68.55; H, 4.65; N, 9.22; S, 7.04. Found: C, 68.74; H, 4.82; N, 9.43; S, 7.18.

*(2E)-3-[4-(Benzyloxy)-3-methoxyphenyl]-2-cyano-N-(3-cyano-4,5,6,7-tetrahydro-1-benzothiophen-2-yl)prop-2-enamide* (**32b**). Compound **32b** was synthesized according to typical procedure using **9b** and 4-(benzyloxy)-3-methoxybenzaldehyde **17** as starting materials. The yield was (0.20 g, 42.38%) as yellow powder with m.p. 220–222 °C. ^1^H NMR (500 MHz, DMSO-*d*_6_) δ (ppm): 1.70–1.85 (4H, m, 2 CH_2_ of C^5,6^-*H*s of tetrahydrobenzothiophene ring), 2.45–2.55 (2H, m, CH_2_ of C^7^-*H*s of tetrahydrobenzothiophene ring), 2.55–2.65 (2H, m, CH_2_ of C^4^-*H*s of tetrahydrobenzothiophene ring), 3.82 (3H, s, CH_3_ of OCH_3_ at C^3^ of phenyl ring), 5.22 (2H, s, benzylic CH_2_), 7.28 (1H, d, *J* = 8.5 Hz, C^5^-*H* of phenyl ring), 7.35 (1H, d, *J* = 6.5 Hz, C^4^-*H* of benzyl group), 7.40 (2H, t, *J* = 6.5 Hz, C^3,5^-*H*s of benzyl group), 7.45 (2H, d, *J* = 6.5 Hz, C^2,6^-*H*s of benzyl group), 7.65 (1H, d, *J* = 8.5 Hz, C^6^-*H* of phenyl ring), 7.74 (1H, s, C^2^-*H* of phenyl ring), 8.25 (1H, s, vinyl-*H*), 11.72 (1H, s, NH). ^13^C NMR (125 MHz, DMSO-*d*_6_) δ (ppm): 21.74, 22.64, 23.57, 23.67, 55.65, 70.10, 112.96, 113.30, 116.82, 116.86, 124.58, 126.00, 128.14 (2C), 128.26, 128.62 (2C), 131.62, 136.31, 149.00, 152.17, 160.78. EI-MS: *m*/*z*: 469.13 [M^+^]. Anal. Calcd. for C_27_H_23_N_3_O_3_S: C, 69.06; H, 4.94; N, 8.95; S, 6.83. Found: C, 69.34; H, 4.81; N, 9.17; S, 7.01.

*(2E)-2-Cyano-N-{3-cyano-4H,5H,6H-cyclopenta[b]thiophen-2-yl}-3-(3-methyl-1-phenyl-1H-pyrazol-4-yl)prop-2-enamide* (**33a**). Compound **33a** was synthesized according to typical procedure using **9a** and 3-methyl-1-phenyl-1*H*-pyrazole-4-carbaldehyde **18** as starting materials. The yield was (0.32 g, 78.8%) as a yellow powder with m.p. 267–269 °C. ^1^H NMR (500 MHz, CDCl_3_) δ (ppm): 1.58 (3H, s, CH_3_ at C^4^ of pyrazole ring), 2.46 (2H, quint, *J* = 7.0 Hz, CH_2_ of C^5^-*H*s of cyclopenta[b]thiophen ring), 2.84 (2H, t, *J* = 7.0 Hz, CH_2_ of C^4^-*H*s of cyclopenta[b]thiophen ring), 2.92 (2H, t, *J* = 7.0 Hz, CH_2_ of C^6^-*H*s of cyclopenta[b]thiophen ring), 7.39 (1H, t, *J* = 8.0 Hz, C^4^-*H* of phenyl ring), 7.51 (2H, t, *J* = 8.0 Hz, 2 CH of C^3,5^-*H* of phenyl ring), 7.72 (2H, d, *J* = 8.0 Hz, 2 CH of C^2,6^-*H* of phenyl ring), 8.34 (1H, s, C5-H of pyrazole ring), 8.80 (1H, s, vinyl-*H*), 8.93 (1H, s, NH). ^13^C NMR (125 MHz, CDCl_3_) δ (ppm): 11.74, 28.06, 28.13, 29.44, 91.28, 97.32, 113.86, 116.11, 116.75, 119.83 (2C), 128.13, 128.76, 129.70 (2C), 136.20, 138.69, 141.58, 145.45, 149.73, 154.23, 157.67. EI-MS: *m*/*z*: 399.8 [M^+^]. Anal. Calcd. for C_22_H_17_N_5_OS: C, 66.15; H, 4.29; N, 17.53; S, 8.03. Found: C, 66.42; H, 4.36; N, 17.79; S, 8.21.

*(2E)-2-Cyano-N-(3-cyano-4,5,6,7-tetrahydro-1-benzothiophen-2-yl)-3-(3-methyl-1-phenyl-1H-pyrazol-4-yl)prop-2-enamide* (**33b**). Compound **33b** was synthesized according to typical procedure using **9b** and 3-methyl-1-phenyl-1*H*-pyrazole-4-carbaldehyde **18** as starting materials. The yield was (0.33 g, 78.59%) as a yellow powder with m.p. 256–258 °C. ^1^H NMR (500 MHz, CDCl_3_) δ (ppm): 1.80–1.90 (4H, m, 2 CH_2_ of C^5,6^-*H*s of tetrahydrobenzothiophene ring), 2.50 (3H, s, CH_3_ at C^4^ of pyrazole ring), 2.62 (2H, t, *J* = 6.0 Hz, CH_2_ of C^7^-*H*s of tetrahydrobenzothiophene ring), 2.67 (2H, t, *J* = 5.5 Hz, CH_2_ of C^4^-*H*s of tetrahydrobenzothiophene ring), 7.39 (1H, t, *J* = 7.5 Hz, C^4^-*H* of phenyl ring), 7.50 (2H, t, *J* = 7.5 Hz, 2 CH of C^3,5^-*H* of phenyl ring), 7.72 (2H, d, *J* = 7.5 Hz, 2 CH of C^2,6^-*H* of phenyl ring), 8.34 (1H, s, C^5^-H of pyrazole ring), 8.81 (1H, s, vinyl-*H*), 8.92 (1H, s, NH). ^13^C NMR (125 MHz, CDCl_3_) δ (ppm): 11.73, 21.99, 22.99, 23.97, 24.01, 95.54, 97.38, 113.60, 116.09, 116.73, 119.79 (2C), 128.10, 128.73, 129.69 (2C), 129.86, 131.65, 138.67, 144.98, 145.45, 154.21, 157.66. EI-MS: *m*/*z*: 413.27 [M^+^]. Anal. Calcd. for C_23_H_19_N_5_OS: C, 66.81; H, 4.63; N, 16.94; S, 7.75. Found: C, 67.09; H, 4.81; N, 17.15; S, 7.94.

*(2E)-2-Cyano-N-{3-cyano-4H,5H,6H-cyclopenta[b]thiophen-2-yl}-3-(3-phenyl-1H-pyrazol-4-yl)prop-2-enamide* (**34a**). Compound **34a** was synthesized according to typical procedure using **9a** and 3-phenyl-1*H*-pyrazole-4-carbaldehyde **19** as starting materials. The yield was (0.24 g, 60.96%) as golden powder with m.p. 226–228 °C. ^1^H NMR (400 MHz, DMSO-*d*_6_) δ (ppm): 2.30–2.50 (2H, m, CH_2_ of C^5^-*H*s of cyclopenta[b]thiophen ring), 2.70–2.95 (4H, m, 2 CH_2_ of C^4,6^-*H*s of cyclopenta[b]thiophen ring), 7.65 (5H, m, 5 CH of C^2,3,4,5,6^-*H*s of phenyl ring), 8.14 (1H, s, vinyl-*H*), 8.56 (1H, br s, NH), 11.71 (1H, s, C^5^-*H* of pyrazole ring), 14.16 (1H, br s, NH of pyrazole ring). ^13^C NMR (100 MHz, DMSO-*d*_6_) δ (ppm): 27.94, 28.10, 29.66, 92.88, 100.33, 112.64, 113.33, 114.86, 117.35, 129.49 (4C), 129.68, 136.00, 142.10 (3C), 151.02, 160.92 (2C). EI-MS: *m*/*z*: 385.48 [M^+^]. Anal. Calcd. for C_21_H_15_N_5_OS: C, 65.44; H, 3.92; N, 18.17; S, 8.32. Found: C, 65.67; H, 4.09; N, 18.30; S, 8.58.

*(2E)-2-Cyano-N-(3-cyano-4,5,6,7-tetrahydro-1-benzothiophen-2-yl)-3-(3-phenyl-1H-pyrazol-4-yl)prop-2-enamide* (**34b**). Compound **34b** was synthesized according to typical procedure using **9b** and 3-phenyl-1*H*-pyrazole-4-carbaldehyde **19** as starting materials. The yield was (0.24 g, 60.07%) as a yellow powder with m.p. 263–265 °C. ^1^H NMR (400 MHz, DMSO-*d*_6_) δ (ppm): 1.70–1.85 (4H, m, 2 CH_2_ of C^5,6^-*H*s of tetrahydrobenzothiophene ring), 2.55–2.65 (4 H, m, 2 CH_2_ of C^4,7^-*H*s of tetrahydrobenzothiophene ring), 7.65 (5H, m, 5 CH of C^2,3,4,5,6^-*H*s of phenyl ring), 8.13 (1H, s, vinyl-*H*), 8.54 (1H, br s, NH), 11.70 (1H, s, C^5^-*H* of pyrazole ring), 14.14 (1H, br s, NH of pyrazole ring). ^13^C NMR (125 MHz, DMSO-*d*_6_) δ (ppm): 21.65, 22.56, 23.47, 23.58, 96.36, 99.89, 112.14, 114.06, 116.86, 129.02 (4C), 129.21, 129.35 (2C), 131.50 (2C), 145.69 (2C), 160.46 (2C). EI-MS: *m*/*z*: 399.22 [M^+^]. Anal. Calcd. for C_22_H_17_N_5_OS: C, 66.15; H, 4.29; N, 17.53; S, 8.03. Found: C, 65.97; H, 4.38; N, 17.71; S, 8.14.

*(2E)-2-Cyano-N-{3-cyano-4H,5H,6H-cyclopenta[b]thiophen-2-yl}-3-(naphthalen-1-yl)prop-2-enamide* (**35a**). Compound **35a** was synthesized according to typical procedure using **9a** and 1-naphthaldehyde **20** as starting materials. The yield was (0.24 g, 64.69%) as a dark orange powder with m.p. 238–240 °C. ^1^H NMR (500 MHz, DMSO-*d*_6_) δ (ppm): 2.38 (2H, quint, *J* = 7.0 Hz, CH_2_ of C^5^-*H*s of cyclopenta[b]thiophen ring), 2.77 (2H, t, *J* = 7.0 Hz, CH_2_ of C^4^-*H*s of cyclopenta[b]thiophen ring), 2.89 (2H, t, *J* = 7.0 Hz, CH_2_ of C^6^-*H*s of cyclopenta[b]thiophen ring), 7.67 (1H, t, *J* = 7.0 Hz, C^6^-*H* of naphthalene ring), 7.70 (2H, t, *J* = 8.0 Hz, 2 CH of C^3,7^-*H*s of naphthalene ring), 8.07 (1H, d, *J* = 8.0 Hz, C^4^-*H* of naphthalene ring), 8.12 (1H, d, *J* = 7.0 Hz, C^5^-*H* of naphthalene ring), 8.18 (1H, d, *J* = 8.0 Hz, C^2^-*H* of naphthalene ring), 8.26 (1H, d, *J* = 8.0 Hz, C^8^-*H* of naphthalene ring), 9.06 (1H, s, vinyl-*H*), 12.06 (1H, s, NH). ^13^C NMR (125 MHz, DMSO-*d*_6_) δ (ppm): 27.50, 27.71, 29.28, 92.48, 108.92, 114.48, 115.73, 124.02, 125.56 (2C), 127.06, 127.72 (2C), 128.90, 129.19, 130.85, 132.59, 133.09, 135.65, 141.91, 151.43, 160.14. EI-MS: *m*/*z*: 369.21 [M^+^]. Anal. Calcd. for C_22_H_15_N_3_OS: C, 71.52; H, 4.09; N, 11.37; S, 8.68. Found: C, 71.38; H, 4.23; N, 11.58; S, 8.90.

*(2E)-2-Cyano-N-(3-cyano-4,5,6,7-tetrahydro-1-benzothiophen-2-yl)-3-(naphthalen-1-yl)prop-2-enamide* (**35b**). Compound **35b** was synthesized according to typical procedure using **9b** and 1-naphthaldehyde **20** as starting materials. The yield was (0.24 g, 63.1%) as a yellow powder with m.p. 265–267 °C. ^1^H NMR (500 MHz, DMSO-*d*_6_) δ (ppm): 1.73–1.83 (4H, m, 2 CH_2_ of C^5,6^-*H*s of tetrahydrobenzothiophene ring), 2.52–2.58 (2H, m, CH_2_ of C^7^-*H*s of tetrahydrobenzothiophene ring), 2.63–2.69 (2H, m, CH_2_ of C^4^-*H*s of tetrahydrobenzothiophene ring), 7.66 (1H, t, *J* = 7.0 Hz, C^6^-*H* of naphthalene ring), 7.70 (2H, t, *J* = 8.0 Hz, 2 CH of C^3,7^-*H*s of naphthalene ring), 8.07 (1H, d, *J* = 8.0 Hz, C^4^-*H* of naphthalene ring), 8.11 (1H, d, *J* = 7.0 Hz, C^5^-*H* of naphthalene ring), 8.18 (1H, d, *J* = 8.0 Hz, C^2^-*H* of naphthalene ring), 8.26 (1H, d, *J* = 8.0 Hz, C^8^-*H* of naphthalene ring), 9.06 (1H, s, vinyl-*H*), 12.06 (1H, s, NH). ^13^C NMR (125 MHz, DMSO-*d*_6_) δ (ppm): 21.67, 22.56, 23.50, 23.63, 96.33, 114.64, 116.25, 124.03, 125.55, 127.05, 127.71, 127.75, 128.89, 129.20, 129.30, 130.83, 132.57, 133.08, 133.17, 133.58, 133.65, 151.51, 160.12. EI-MS: *m*/*z*: 383.23 [M^+^]. Anal. Calcd. for C_23_H_17_N_3_OS: C, 72.04; H, 4.47; N, 10.96; S, 8.36. Found: C, 71.88; H, 4.69; N, 11.23; S, 8.49.

*(2E)-2-Cyano-N-{3-cyano-4H,5H,6H-cyclopenta[b]thiophen-2-yl}-3-(naphthalen-2-yl)prop-2-enamide* (**36a**). Compound **36a** was synthesized according to typical procedure using **9a** and 2-naphthaldehyde **21** as starting materials. The yield was (0.19 g, 51.42%) as a dark orange powder with m.p. 228–230 °C. ^1^H NMR (500 MHz, DMSO-*d*_6_) δ (ppm): 2.25–2.35 (2H, m, CH_2_ of C^5^-*H*s of cyclopenta[b]thiophen ring), 2.65–2.80 (4H, m, 2 CH_2_ of C^4,6^-*H*s of cyclopenta[b]thiophen ring), 7.54–7.66 (2H, m, C^6,7^-*Hs* of naphthalene ring), 7.92–8.02 (3H, m, C^3,5,8^-*Hs* of naphthalene ring), 8.15 (1H, d, *J* = 8.0 Hz, C^4^-*H* of naphthalene ring), 8.38 (1H, s, C^1^-*H* of naphthalene ring), 8.38 (1H, s, vinyl-*H*). ^13^C NMR (125 MHz, DMSO-*d*_6_) δ (ppm): 27.06, 27.95, 29.63, 87.54, 113.98, 118.68 (2C), 125.09, 126.97, 127.72, 127.83, 128.55, 128.76, 130.00 (2C), 131.03, 131.09, 132.75, 134.00, 139.05, 146.09, 162.53. EI-MS: *m*/*z*: 369.53 [M^+^]. Anal. Calcd. for C_22_H_15_N_3_OS: C, 71.52; H, 4.09; N, 11.37; S, 8.68. Found: C, 71.69; H, 4.21; N, 11.62; S, 8.79.

*(2E)-2-Cyano-N-(3-cyano-4,5,6,7-tetrahydro-1-benzothiophen-2-yl)-3-(naphthalen-2-yl)prop-2-enamide* (**36b**). Compound **36b** was synthesized according to typical procedure using **9b** and 2-naphthaldehyde **21** as starting materials. The yield was (0.19 g, 49.5%) as a dark yellow powder with m.p. 252–254 °C. ^1^H NMR (500 MHz, DMSO-*d*_6_) δ (ppm): 1.73–1.81 (4H, m, 2 CH_2_ of C^5,6^-*H*s of tetrahydrobenzothiophene ring), 2.50–2.55 (2H, m, CH_2_ of C^7^-*H*s of tetrahydrobenzothiophene ring), 2.59–2.65 (2H, m, CH_2_ of C^4^-*H*s of tetrahydrobenzothiophene ring), 7.63 (1H, t, *J* = 7.5 Hz, C^6^-*H* of naphthalene ring), 7.68 (1H, t, *J* = 7.5 Hz, C^7^-*H* of naphthalene ring), 8.01 (1H, d, *J* = 8.0 Hz, C^5^-*H* of naphthalene ring), 8.06 (1H, d, *J* = 8.0 Hz, C^8^-*H* of naphthalene ring), 8.10 (1H, d, *J* = 9.0 Hz, C^3^-*H* of naphthalene ring), 8.19 (1H, d, *J* = 9.0 Hz, C^4^-*H* of naphthalene ring), 8.44 (1H, s,C^1^-*H* of naphthalene ring), 8.51 (1H, s, vinyl-*H*), 11.94 (1H, s, NH). ^13^C NMR (125 MHz, CF_3_COOD) δ (ppm): 23.03, 25.17, 25.73, 26.23, 114.74, 115.16, 124.92, 125.31, 129.68, 129.80, 130.02, 130.16, 130.90, 131.55, 131.88, 134.64, 135.02, 135.96, 136.88, 146.66, 151.79, 158.56, 162.32. EI-MS: *m*/*z*: 383.26 [M^+^]. Anal. Calcd. for C_23_H_17_N_3_OS: C, 72.04; H, 4.47; N, 10.96; S, 8.36. Found: C, 71.92; H, 4.65; N, 11.18; S, 8.47.

*(2E)-3-(2H-1,3-Benzodioxol-4-yl)-2-cyano-N-{3-cyano-4H,5H,6H-cyclopenta[b]thiophen-2-yl}prop-2-enamide* (**37a**). Compound **37a** was synthesized according to typical procedure using **9a** and 2,3-methylenedioxybenzaldehyde **22** as starting materials. The yield was (0.30 g, 82.55%) as a yellow powder with m.p. 286–288 °C. ^1^H NMR (500 MHz, DMSO-*d*_6_) δ (ppm): 2.36 (2H, quint, *J* = 7.0 Hz, CH_2_ of C^5^-*H*s of cyclopenta[b]thiophen ring), 2.75 (2H, t, *J* = 7.0 Hz, CH_2_ of C^4^-*H*s of cyclopenta[b]thiophen ring), 2.86 (2H, t, *J* = 7.0 Hz, CH_2_ of C^6^-*H*s of cyclopenta[b]thiophen ring), 6.21 (2H, s, CH_2_ of C^2^-*H*s of benzodioxole ring), 7.04 (1H, t, *J* = 7.75 Hz, C^6^-*H* of benzodioxole ring), 7.16 (1H, d, *J* = 7.75 Hz, C^5^-*H* of benzodioxole), 7.63 (1H, d, *J* = 7.75 Hz, C^7^-*H* of benzodioxole ring), 8.23 (1H, s, vinyl-*H*), 11.91 (1H, s, NH). ^13^C NMR (125 MHz, CF_3_COOD) δ (ppm): 29.46, 29.59, 30.98, 100.17, 104.41, 110.21, 115.87, 116.37, 117.46, 121.20, 124.79, 140.53, 149.70, 150.43, 151.23, 152.48, 152.65, 163.39, 171.42. EI-MS: *m*/*z*: 363.33 [M^+^]. Anal. Calcd. for C_19_H_13_N_3_O_3_S: C, 62.80; H, 3.61; N, 11.56; S, 8.82. Found: C, 63.04; H, 3.78; N, 11.82; S, 8.98.

*(2E)-3-(2H-1,3-Benzodioxol-4-yl)-2-cyano-N-(3-cyano-4,5,6,7-tetrahydro-1-benzothiophen-2-yl)prop-2-enamide* (**37b**). Compound **37b** was synthesized according to typical procedure using **9b** and 2,3-methylenedioxybenzaldehyde **22** as starting materials. The yield was (0.20 g, 52.99%) as a yellow powder with m.p. 226–228 °C. ^1^H NMR (500 MHz, CDCl_3_) δ (ppm): 1.80–1.90 (4H, m, 2 CH_2_ of C^5,6^-*H*s of tetrahydrobenzothiophene ring), 2.60–2.70 (4H, m, 2 CH_2_ of C^4,7^-*H*s of tetrahydrobenzothiophene ring), 6.10 (2H, s, CH_2_ of C^2^-*H*s of benzodioxole ring), 6.93–6.99 (2H, m, C^5,6^-*H*s of benzodioxole ring), 6.75 (1H, d, *J* = 7.5 Hz, C^7^-*H* of benzodioxole ring), 8.55 (1H, s, vinyl-H), 8.93 (1H, br s, NH). ^13^C NMR (100 MHz, CDCl_3_) δ (ppm): 22.03, 23.03, 24.02, 24.08, 95.97, 102.00, 102.16, 113.02, 113.54, 114.24, 115.98, 119.78, 122.52, 130.17, 131.81, 144.78, 147.02, 148.21, 149.58, 157.11. EI-MS: *m*/*z*: 377.13 [M^+^]. Anal. Calcd. for C_20_H_15_N_3_O_3_S: C, 63.65; H, 4.01; N, 11.13; S, 8.49. Found: C, 63.89; H, 4.23; N, 11.41; S, 8.60.

*(2E)-3-(2H-1,3-Benzodioxol-5-yl)-2-cyano-N-{3-cyano-4H,5H,6H-cyclopenta[b]thiophen-2-yl}prop-2-enamide* (**38a**). Compound **38a** was synthesized according to typical procedure using **9a** and 3,4-methylenedioxybenzaldehyde **23** as starting materials. The yield was (0.22 g, 60.54%) as a golden powder with m.p. 219–221 °C. ^1^H NMR (500 MHz, DMSO-*d*_6_) δ (ppm): 2.25–2.35 (2H, m, CH_2_ of C^5^-*H*s of cyclopenta[b]thiophen ring), 2.60–2.80 (4H, m, 2 CH_2_ of C^4,6^-*H*s of cyclopenta[b]thiophen ring), 6.19 (2H, s, CH_2_ of C^2^-*H*s of benzodioxole ring), 7.15 (1H, d, *J* = 10.0 Hz, C^7^-*H* of benzodioxole ring), 7.58 (1H, d, *J* = 10.0 Hz, C^6^-*H* of benzodioxole ring), 7.65 (1H, s, C^4^-*H* of benzodioxole), 8.24 (1H, s, vinyl-*H*), 11.73 (1H, s, NH). ^13^C NMR (125 MHz, DMSO-*d*_6_) δ (ppm): 27.53, 27.73, 29.28, 92.59, 101.39, 102.64, 108.19, 109.26, 114.45, 116.49, 129.09, 135.71, 141.87, 148.27, 150.47, 151.83, 152.86, 160.55. EI-MS: *m*/*z*: 363.13 [M^+^]. Anal. Calcd. for C_19_H_13_N_3_O_3_S: C, 62.80; H, 3.61; N, 11.56; S, 8.82. Found: C, 63.07; H, 3.49; N, 11.80; S, 8.65.

*(2E)-3-(2H-1,3-Benzodioxol-5-yl)-2-cyano-N-(3-cyano-4,5,6,7-tetrahydro-1-benzothiophen-2-yl)prop-2-enamide* (**38b**). Compound **38b** was synthesized according to typical procedure using **9b** and 3,4-methylenedioxybenzaldehyde **23** as starting materials. The yield was (0.20 g, 52.99%) as a yellow powder with m.p. 204–206 °C. FT-IR (KBr) *ν*_max_ (cm^−1^): 1279.83 (C-O of benzodioxole ring), 1648.26 (C=O of amide), 1673.84 (C=C), 2223.38 (CN), 2931.53 (aliphatic protons), 3002.25 (aromatic protons), 3433.83 (NH of amide). ^1^H NMR (400 MHz, CDCl_3_) δ (ppm): 1.80–1.90 (4H, m, 2 CH_2_ of C^5,6^-*H*s of tetrahydrobenzothiophene ring), 2.60–2.75 (4H, m, 2 CH_2_ of C^4,7^-*H*s of tetrahydrobenzothiophene ring), 6.14 (2H, s, CH_2_ of C^2^-*H*s of benzodioxole ring), 6.96 (1H, d, *J* = 8.0 Hz, C^7^-*H* of benzodioxole ring), 7.47 (1H, d, *J* = 8.0 Hz, C^6^-*H* of benzodioxole ring), 7.71 (1H, s, C^4^-*H* of benzodioxole ring), 8.30 (1H, s, vinyl-*H*), 8.96 (1H, s, NH). NOESY: dipolar interaction detected between C^6^-*H* of benzodioxole ring (7.47, d) and vinyl-*H* (8.30, s) and between C^4^-*H* of benzodioxole ring (7.71, s) and vinyl-*H* (8.30, s). ^13^C NMR (100 MHz, CDCl_3_) δ (ppm): 17.30, 18.30, 19.28, 19.33, 91.04, 93.89, 97.76, 104.07, 104.39, 108.82, 111.69, 121.20, 125.24, 125.60, 127.01, 140.22, 144.12, 148.12, 149.74, 152.96. EI-MS: *m*/*z*: 377.53 [M^+^]. Anal. Calcd. for C_20_H_15_N_3_O_3_S: C, 63.65; H, 4.01; N, 11.13; S, 8.49. Found: C, 63.81; H, 4.23; N, 11.40; S, 8.65.

*(2E)-2-Cyano-N-{3-cyano-4H,5H,6H-cyclopenta[b]thiophen-2-yl}-3-(1H-indol-3-yl)prop-2-enamide* (**39a**). Compound **39a** was synthesized according to typical procedure using **9a** and indole-3-carboxaldehyde **24** as starting materials. The yield was (0.23 g, 65.28%) as a golden powder with m.p. 284–286 °C. ^1^H NMR (500 MHz, DMSO-*d*_6_) δ (ppm): 2.30–2.40 (2H, m, CH_2_ of C^5^-*H*s of cyclopenta[b]thiophen ring), 2.70–2.95 (4H, m, 2 CH_2_ of C^4,6^-*H*s of cyclopenta[b]thiophen ring), 7.25–7.35 (2H, m, 2 CH of C^5,6^-*H*s of indole ring), 7.57 (1H, d, *J* = 6.5 Hz, C^7^-*H* of indole ring), 7.99 (1H, d, *J* = 5.5 Hz, C^4^-*H* of indole ring), 8.57 (1H, s, C^4^-*H* of indole ring), 8.64 (1H, s, vinyl-*H*), 11.52 (1H, s, NH), 12.54 (1H, s, NH of indole ring). ^13^C NMR (100 MHz, DMSO-*d*_6_) δ (ppm): 27.92, 28.13, 29.66, 92.49, 95.20, 110.47, 113.44, 115.06, 118.74, 119.10, 122.51, 124.11, 127.62, 132.39, 135.61, 136.65, 142.03, 145.62, 151.53, 161.66. EI-MS: *m*/*z*: 358.42 [M^+^]. Anal. Calcd. for C_20_H_14_N_4_OS: C, 67.02; H, 3.94; N, 15.63; S, 8.94. Found: C, 66.85; H, 4.12; N, 15.79; S, 9.08.

*(2E)-2-Cyano-N-(3-cyano-4,5,6,7-tetrahydro-1-benzothiophen-2-yl)-3-(1H-indol-3-yl)prop-2-enamide* (**39b**). Compound **39b** was synthesized according to typical procedure using **9b** and indole-3-carboxaldehyde **24** as starting materials. The yield was (0.20 g, 53.69%) as dark yellow powder with m.p. 284–286 °C. ^1^H NMR (400 MHz, DMSO-*d*_6_) δ (ppm): 1.60–1.90 (4H, m, 2 CH_2_ of C^5,6^-*H*s of tetrahydrobenzothiophene ring), 2.60–2.70 (4 H, m, 2 CH_2_ of C^4,7^-*H*s of tetrahydrobenzothiophene ring), 7.20–7.40 (2H, m, 2 CH of C^5,6^-*H*s of indole ring), 7.61 (1H, s, C^7^-*H* of indole ring), 8.02 (1H, s, C^4^-*H* of indole ring), 8.61 (1H, s, C^2^-*H* of indole ring), 8.67 (1H, s, vinyl-*H*), 11.57 (1H, s, NH), 12.58 (1H, s, NH of indole ring). ^13^C NMR (100 MHz, CDCl_3_) δ (ppm): 22.17, 23.07, 23.95, 24.09, 31.19, 95.28, 110.46, 113.46, 114.74, 118.76, 119.15, 122.51, 124.11, 127.61, 129.33, 131.91, 132.43, 136.67, 145.70, 146.61, 161.69. EI-MS: *m*/*z*: 372.18 [M^+^]. Anal. Calcd. for C_21_H_16_N_4_OS: C, 67.72; H, 4.33; N, 15.04; S, 8.61. Found: C, 67.51; H, 4.60; N, 15.28; S, 8.73.

### 3.2. In Vitro Telomerase Activity Assay

The tested compound and **BIBR1532** (R&D Systems, Minneapolis, MN, USA) were dissolved in an appropriate solvent (according to recommendations from TSCA certification) at stock concentrations of 10 mM, and further diluted to appropriate concentrations in assay buffer. The Telomeric Repeat Amplification Protocol (TRAP) [61] was used to evaluate the activity of telomerase, with some changes that we have previously described [62,63]. A549 cells (human epithelial cell lung carcinoma, ATCC, Manassas, VA) were lysed in (10 mM) Tris-HCl, pH 7.5, (1 mM) EGTA, (1 mM) MgCl_2_, (5 mM) 2-mercaptoethanol, (0.1 mM) PMSF, (0.5%) CHAPS, and (10%) glycerol (Sigma-Aldrich, St. Louis, MO, USA) and centrifugation. The supernatants were kept at −80 °C. The BCA-1 Protein Assay Kit was used to assess the protein concentration in cell extracts (Sigma-Aldrich, St. Louis, MO, USA). An amount of 5 µg total protein and appropriate amounts of AZT and **BIBR1532** were added to a reaction mixture of (67 mM) Tris-HCl, pH 8.8, (1.5 mM) MgCl_2_, (0.01%) Tween-20, (16.6 mM) (NH_4_)_2_SO_4_, and (1 mM) EGTA (Sigma-Aldrich, St. Louis, MO, USA), (0.25 mM) each dNTP (Evrogen, Moscow) (TS-primer) (5′-AATCCGTCGAGCAGAGTT-3′). To inactivate telomerase, the cells were elongated for 30 min at 37 °C and 10 min at 96 °C. Copy CX-primer was added to the elongation mixture along with 0.1 µL (5′-CCCTTACCCTTACCCTTACCCTAA-3′) and 2.5 units Taq-polymerase, followed by the following PCR: 94 °C for 5 min; 30 cycles of 94 °C for 30 s, 50 °C for 30 s, and 72 °C for 40 s, and 72 °C for 5 min. TBE buffer and (12%) nondenaturing PAAG electrophoresis were used to visualize the PCR product. 10 µL of each sample was placed in each well of gel comb. The gels were stained with the SYBR Green I (Invitrogen, Grand Island, NY, USA), then photographed under the UV light in ChemiDocTM XRS imaging system and analyzed with GelAnalyzer 2010.

### 3.3. Cell Lines and Incubation with the Compounds

The human A549 (epithelial cell lung carcinoma, ATCC, Manassas, VA, USA), HCC44 (non-small cell lung adenocarcinoma, Leibniz Institute DSMZ-German Collection of the Microorganisms and the Cell Cultures, Braunschweig, Germany), and NCI-H23 (non-small cell lung adenocarcinoma, ATCC, Manassas, VA, USA) cell lines (compounds **29a**, **36b**, and **39b**) were diluted to 10 µM and incubated for 48 h before being tested using the TRAP method.

### 3.4. In Vitro Anticancer Screening

The cancer screening panel’s human tumor 60 cell lines were cultured in the RPMI 1640 medium with (5%) fetal bovine serum and (2 mM) L-glutamine. For a screening experiment, cells were injected onto 96-well microtiter plates (100 µL), with plating densities varying from 5 to 40 × 10^3^ cells per well, depending on different cell lines’ doubling times. Before introducing experimental drugs, the microtiter plates were incubated overnight at 37 °C, 95% air, 5% CO_2_, and 100% relative humidity after cell injection. TCA was employed to keep each cell line in place overnight, resembling a computation of the cell population for every cell line at the time of the addition of the drug. The experimental drugs were chilled after solubilization in DMSO at 400 times the final maximum test concentration. At the moment of the addition of the drug, the aliquot of the frozen concentrate was thawed and diluted to twice the required final maximum concentration with the complete medium containing (50 µg/mL) gentamicin. The required final concentration of the drug (10 µM) was reached by adding 100 µL aliquots of these definite dilutions of the drug to suitable microtiter wells previously holding a medium of 100 µL. The same technique was utilized for controls containing only DMSO and phosphate-buffered saline at identical dilutions. Following drug addition, the plates were incubated for a further two days at 37 °C, 95% air, 5% CO_2_, and 100% relative humidity. The experiment was finished with addition of cold TCA. The cells were fixed in situ by adding (50 µL) cold 50% (*w*/*v*) TCA (final concentration: 10% TCA) and incubating for 1 hr at 4 °C. The plates were rinsed numerous times with distilled water and dried after the supernatant was removed. Sulforhodamine B (SRB) solution (100 µL) containing 0.4% (*w*/*v*) sulforhodamine in 1% acetic acid was added and plates were incubated at rt for 10 min. The unbound dye was washed away numerous times with (1%) acetic acid after the staining, and plates were dried. The bound dye was then solubilized with a (10 mM) trizma base, and absorbance was calculated at 515 nm on an automatic reader. The process was typical for suspension cells, except that the experiment was finished by gradually adding (50 µL) of 80% TCA to fix settling cells at the bottoms of the wells (final conccentration:, 16% TCA). The % growth of treated cells was established using seven absorbance measurements and compared to untreated control cells [57].

### 3.5. Statistical Analysis

Analysis was carried out using SPSS 25 software, a 2-way ANOVA, and a student’s *t*-test (IBM SPSS Statistics, Armonk, NY, USA). The mean ± SEM is used to express the findings. The significance level was set at *p* ≤ 0.05. An amount of 1 µL was submitted to the Real-Time Quantitative Telomeric Repeat Amplification Protocol Assay (RTQ-TRAP) as reported by Hou M., et al. to obtain the IC_50_ and IC_90_ values (inhibitor concentrations where the response is decreased by 50% and 90%, respectively) [64]. According to Sebaugh J.L., et al.’s guidelines, the values were assessed using the Prism 6 software (GraphPad, San Diego, CA, USA) [65].

### 3.6. Molecular Docking

The RSCB Protein Data Bank was employed to get the structure of telomerase from *Tribolium castaneum* (PDB ID: 5CQG) for complex modelling [66]. The SYBYL 8.1 suite was used to design the structures of compound **36b**. The structure was optimized in a vacuum using a Tripos force field and energy minimization. The Gasteiger–Huckel method was used to compute the partial atomic charges. The AutoDock Vina package was used to perform the docking [67]. The docking parameters were established using the AutoDock Tools package. Based on the values of their scoring functions and poses in the binding site, the ligand poses acquired via docking were graded and chosen. Ligand positions from crystal structures were used as a reference template to assess the accuracy of the docked molecules’ poses. The PLIP server and the PyMol package were used to study intermolecular interactions between proteins and docked molecules [68].

### 3.7. Molecular Dynamics Simulation

The Gromacs-2020 software package was used to simulate molecular dynamics, with the explicit solvent (TIP3P) and Na+ ions used to neutralize the system. For atomic parametrization of protein molecules, the AMBER99SB-ILDN forcefield was employed, while for ligand molecules, the GAFF forcefield was used. Steepest descent minimization with a solvent was performed for 50,000 steps. The minimization was followed by a 5 ns NVT equilibration, which was followed by a 5 ns NPT equilibration. During both phases of equilibration, movements of protein- and ligand-heavy atoms were restricted. The equilibrated structure was used as a starting point for MD. MD calculations were performed during 100 ns trajectories. An integrator step was set to 2 fs. The values of the temperature and the pressure were set to 300 K and 1 atm, respectively. The Berendsen thermostat was used for temperature coupling, and the Parrinelo–Ranman barostat was used for pressure coupling. Hydrogen atoms were restricted with the LINCS algorithm. The particle mesh Ewald (PME) was utilized to treat long-range electrostatic interactions. The cutoff distance for nonbonded interactions was set to 12 Å. Data frames were saved every 10 ps. To avoid nonequilibrium effects, the last 50 ns were used for analysis. For simulations, the docked pose of compound **36b** was employed as a starting point. The Gromacs-2020 built-in tools and VMD-1.9.1 software were used to perform trajectory analysis. The conformational changes in the interaction were identified using RMSD of protein structures and RMSF of residues.

### 3.8. In Silico Pharmacokinetic; Physicochemical Prediction and PAINS Filters

The Swiss Institute of Bioinformatics (SIB) provided the free SwissADME web tool for calculating physicochemical parameters, pharmacokinetic properties, drug-like nature, and medicinal chemistry friendliness [58,59,69]. The structures of **BIBR1532** and compound **36b** were translated to the SMILES data and then uploaded to the online server for evaluation.

## 4. Conclusions

Our goal was to design a new, simply synthesized, and highly derivable chemical scaffold as a telomerase inhibitor. Therefore, 30 novel compounds were designed and simply synthesized using readily available and inexpensive starting materials. The activity of telomerase of all synthesized compounds was assessed through TRAP assay. Compounds **29a**, **36b**, and **39b** showed the greatest inhibitory effect on the telomerase enzyme. The most active compound was **36b**, with IC_50_ values of 0.3 μM. The IC_50_ of compounds **29a** and **39b** were 1.7 and 2 μM, respectively. To test whether these compounds (**29a**, **36b**, and **39b**) can penetrate the cells, a living-cell TRAP assay was performed using three NSCLC cell lines: A549, HCC44, and NCI-H23. All three compounds were successfully capable of penetrating the cell membrane. Compound **36b** was selected to investigate whether it can bind to another target using the NCI-60 cell line panel assay as a screening test. Surprisingly, the growth percentage of the 60 cell lines was barely affected, confirming the selectivity of compound **36b** along with the PAINS filter. According to SwissADME prediction, compound **36b** has a good CNS safety profile and enhanced bioavailability in comparison with **BIBR1532**. The simple synthesis, easily modifiable structure, and cellular penetration capability offer our new scaffold as a valuable new genetic bomb trigger.

## Data Availability

Data is contained within the article and Appendix A.

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
