# Peer review of "New Genetic Bomb Trigger: Design, Synthesis, Molecular Dynamics Simulation, and Biological Evaluation of Novel BIBR1532-Related Analogs Targeting Telomerase against Non-Small Cell Lung Cancer"

_pharmaceuticals, 2022, doi:10.3390/ph15040481_

Round 1
Reviewer 1 Report
The reviewed manuscript describes the synthesis and activity of a 30 compounds with 2-cyano-N-(3-cyano-benzo[b]thiophen-2-yl)-acetamide core as the analogs of BIBR1532. All derivatives have been evaluated as potential telomerase inhibitor. The chemistry is known, however it is worth noting that described derivatives are new compounds. One of the most active derivative (36b) exhibited activity very similar to the BIBR1532 used as standard. Additionally the most active 36b was tested towards 60 cancer cell lines. In my opinion, the biological part is the most interesting part of the presented results. Therefore, reviewed manuscript present new and interesting biological results which provide a good starting point for further research in this area.
I would recommend the publication of reviewed manuscript in Pharmaceuticals after some minor revisions according to the following comments:
- Anticancer activity of the compounds with 2-cyano-N-(3-cyano-benzo[b]thiophen-2-yl)-acetamide core in the structure are known in the literature. Thus, the point 2.1. have to be supplemented with appropriate references (10.2174/1871520617666170110154110; 10.3390/molecules16010052 and 10.1007/s00044-016-1654-3)
- Page 6/lines 158-161: authors assumed that the main isomer of the obtained final products has E configuration, however there is no evidence that confirms it. (cited reference 49 applies different group of Knoevenagel products)
- Page 16/lines 413-416: why was the 1HNMR spectrum of 9b performed in CDCl3, and not in DMSO-d6 as for 9a? Especially that the copy of 1HNMR of 9b (SI file, page 2) clearly indicated that solubility of 9b in CDCl3 is very low (generally, quality of this spectrum is low). Furthermore integration of the signal at 1.8-2.1ppm is 9.9H, not 4H as it state at page 16/line 414.
- Page 16/line 421: K2CO3 is just a base not a catalyst.
- Some 13C NMR spectra placed in SI file are very low quality (signal to noise ratio is very low) and what’s more their analysis is overinterpreted. For example in the case of compound 25a the signal at 192.48 ppm (page 16/line 449) is practically invisible on the copy of the original spectrum (SI file of page 3). In my opinion you have two possibility (in the case of similar "problematic" 13C NMR spectra: 1) repeat the spectrum/spectra with much longer acquisition time (until appropriate signal appear); 2) you can list all clearly indicated signals and add the note/information that there is lack of some signal/s due to very low solubility of the compound.
Reviewer 2 Report
This manuscript describes a series of research on the novel telomerase inhibitors. Base on the potent telomerase inhibitor BIBR1532’s pharmacophoric features, the authors design and synthesize novel BIBR1532-related analogous as inhibitors of telomerase enzyme. Accordingly, three series of compounds, total thirty, were synthesized and biologically evaluated as telomerase inhibitors using the TRAP assay. Compound 36b, with the greatest inhibitory effect, was screened for its cytotoxic activity toward 60 cancer cell lines, suggesting selectivity for the telomerase enzyme. In addition, molecular docking and molecular dynamics simulation for compound 36b were performed to study the interaction with the telomerase allosteric binding site.
The manuscript has substantial contents. The results developed in this manuscript should be of great interest to medical personnel. I recommend this paper to be published in PHAMACEUTICALS.
The concerns that I have are as followings:
- Although the configurations of the designed compounds and BIBR1532 are same as E-form, there are great differences in the structure of their space substitution, such as the -CN group. Can such differences be shown in the molecular docking diagram, such as the interaction with specific amino acid residues?
- What does the description “Genetic Bomb Trigger” in the title refer to? There is no information available about this concept in the full text.
